# Fecal Microbiota Transplantation Prevents Intestinal Injury, Upregulation of Toll-Like Receptors, and 5-Fluorouracil/Oxaliplatin-Induced Toxicity in Colorectal Cancer

**DOI:** 10.3390/ijms21020386

**Published:** 2020-01-08

**Authors:** Ching-Wei Chang, Hung-Chang Lee, Li-Hui Li, Jen-Shiu Chiang Chiau, Tsang-En Wang, Wei-Hung Chuang, Ming-Jen Chen, Horng-Yuan Wang, Shou-Chuan Shih, Chia-Yuan Liu, Tung-Hu Tsai, Yu-Jen Chen

**Affiliations:** 1Institute of Traditional Medicine, National Yang-Ming University, Taipei 11221, Taiwan; wei591026@gmail.com; 2Division of Gastroenterology, Department of Internal Medicine, MacKay Memorial Hospital, Taipei 10449, Taiwan; tsangen.wang@gmail.com (T.-E.W.); mingjen.ch@msa.hinet.net (M.-J.C.); mmh4013@gmail.com (H.-Y.W.); shihshou@gmail.com (S.-C.S.); 3Department of Medical Research, MacKay Memorial Hospital, New Taipei City 25173, Taiwan; 8231boss@gmail.com (H.-C.L.); lihua.530@yahoo.com.tw (L.-H.L.); chiang1997@hotmail.com (J.-S.C.C.); 4Department of Medicine, MacKay Medical College, New Taipei City 25245, Taiwan; 5MacKay Junior College of Medicine, Nursing, and Management, New Taipei City 11260, Taiwan; 6MacKay Children’s Hospital, Taipei 10449, Taiwan; 7Institute of Biomedical Informatics, Center for Systems and Synthetic Biology, National Yang-Ming University, Taipei 11221, Taiwan; hilelojack@gmail.com; 8Department of Chemical Engineering, National United University, Miaoli 36063, Taiwan; 9Department of Radiation Oncology, MacKay Memorial Hospital, Taipei 10449, Taiwan

**Keywords:** FOLFOX, fecal microbiota transplant, intestinal mucositis, apoptosis, gut microbiota

## Abstract

FOLFOX (5-fluorouracil, leucovorin, and oxaliplatin), a 5-fluorouracil (5-FU)-based chemotherapy regimen, is one of most common therapeutic regimens for colorectal cancer. However, intestinal mucositis is a common adverse effect for which no effective preventive strategies exist. Moreover, the efficacy and the safety of fecal microbiota transplants (FMT) in cancer patients treated with anti-neoplastic agents are still scant. We investigated the effect of FMT on FOLFOX-induced mucosal injury. BALB/c mice implanted with syngeneic CT26 colorectal adenocarcinoma cells were orally administered FMT daily during and two days after five-day injection of FOLFOX regimen for seven days. Administration of FOLFOX significantly induced marked levels of diarrhea and intestinal injury. FMT reduced the severity of diarrhea and intestinal mucositis. Additionally, the number of goblet cells and zonula occludens-1 decreased, while apoptotic and NF-κB-positive cells increased following FOLFOX treatment. The expression of toll-like receptors (TLRs), MyD88, and serum IL-6 were upregulated following FOLFOX treatment. These responses were attenuated following FMT. The disrupted fecal gut microbiota composition was also restored by FMT after FOLFOX treatment. Importantly, FMT did not cause bacteremia and safely alleviated FOLFOX-induced intestinal mucositis in colorectal cancer-bearing mice. The putative mechanism may involve the gut microbiota TLR-MyD88-NF-κB signaling pathway in mice with implanted colorectal carcinoma cells.

## 1. Introduction

The microbiota formed by microorganisms and residing in the gastrointestinal tract is referred to as “intestinal microbiota” or “gut microbiota” [1,2]. The microbiota affects various aspects of human health, including providing nutrients and vitamins, protecting against pathogens, epithelial mucosa homeostasis, and immune system development [3]. Microbial dysbiosis has been linked to various metabolic and inflammatory diseases, such as diabetes mellitus, hypertension, inflammatory bowel disease, and obesity [1,2]. Growing evidence not only implies that chemotherapeutics affect the intestinal microbial composition but also that multidirectional interactions between the gut microbiota and the host immune system may influence development and progression of chemotherapy-induced intestinal inflammation [4,5,6].

Gastrointestinal toxicity is a severe, dose-limiting, toxic side effect of chemotherapeutics in cancer patients. Mucositis and diarrhea are a primary outcome of intestinal toxicity and result in an increased hospitalization duration and infection risk as well as reduced anti-neoplastic treatment efficacy, leading to reduced survival and a substantial burden on Medicare [7,8]. The development of chemotherapy-induced mucositis involves a complex and dynamic array of biological events occurring in five interconnected phases, i.e., initiation, upregulation and generation of messenger signals, signal amplification, ulceration, and healing [7,9]. Several pathogenic elements, including direct toxicity, change in the bowel microbial flora, oxidative stress, apoptosis, hypo-proliferation, and abnormal inflammation, are involved. Unfortunately, no well-established or effective therapeutic strategies are currently available for the management of chemotherapy-induced intestinal mucositis [8].

Fecal microbiota transplant (FMT) is the transfer of fecal material, including bacteria and natural anti-bacterials, from a healthy individual into a diseased recipient. The manipulation of gut microbiota by FMT has been performed in various disease models and clinical trials and has received much public attention [10,11]. Due to the high cure rate (over 90%) and the rarity of side effects associated with it, FMT has been considered a potentially life-saving, “last chance” option for the treatment of recurrent *Clostridium difficile* infections [12]. This has generated interest in its use for the treatment of other gastrointestinal and even inflammatory diseases. Several studies have reported that the infusion of a fecal suspension from healthy individuals to patients with inflammatory bowel disease or irritable bowel syndrome resulted in clinical improvement and remission [12,13]. It has been also suggested that FMT shows promising therapeutic potential for the treatment of diabetes, obesity, non-alcoholic fatty liver disease, and even cardiovascular disease [12,13]. Although the underlying molecular and cellular mechanism for FMT remains unclear, it may involve the direct interaction of donor gut microbiota with that of the host, which subsequently mediates the effects observed on host physiology, gut mucosal barrier, and immune system [14]. Thus, FMT may be effective at manipulating gut microbiota and ameliorating inflammation and severity of mucositis induced by chemotherapy.

Several concerns have been raised regarding the safety of FMT therapy. Specifically, adverse events occur often and should be carefully monitored throughout the FMT process [15]. In theory, it may be possible to transmit the potentially harmful traits of microbiota, thereby enabling the transmission of occult infections [16,17]. Further, transient adverse events, including mild fever, abdominal pain, diarrhea, exhaustion, flatulence, and fatigue, were reported following FMT [16,17]. These adverse effects were self-limiting. FMT was proven safe in the short-term follow-up period; however, long-term safety must be assessed [16,17]. Furthermore, though evidence regarding the efficacy and the safety of FMT in immunocompromised patients has begun to emerge, data pertaining to cancer patients treated with anti-neoplastic agents remain scarcely available [18,19,20].

Regimens based on 5-fluorouracil (5-FU) with the cytotoxic agent oxaliplatin, FOLFOX (5-fluorouracil, leucovorin, and oxaliplatin) have been widely used in standard chemotherapy for advanced and metastatic colorectal cancer treatment [21,22]. The combination therapy of oxaliplatin with 5-FU potentiates gastrointestinal toxicity in clinical studies; however, the underlying mechanism remains unclear [7,23]. In our previous study, we reported on a colorectal cancer murine model with FOLFOX-induced intestinal mucositis [24]. It involves changes in gut microbiota and may be “driven” by NF-κB expression, which subsequently induces the generation of apoptotic signals and pro-inflammatory cytokines, which contribute to gastrointestinal injury [24]. By modulating the gut microbiota and the pro-inflammatory responses, probiotics mitigated FOLFOX-induced mucositis [24].

Accordingly, in the current study, we sought to investigate the effect and the safety of FMT on 5-FU-based chemotherapy (FOLFOX)-induced intestinal mucosal injury in mice with colon cancer. We performed FMT using stool from healthy wild-type donor mice to restore microbial diversity and composition. The possible mechanisms of action for FMT were also elucidated.

## 2. Results

### 2.1. The Effects and the Safety of FMT on Colorectal Cancer-Implanted Mice Challenged with FOLFOX

Mice were divided into six treatment groups (Table 1), and experimental animals received 200 μL per day of fecal slurry for FMT (FMT50: 50 mg/mL or FMT150: 150 mg/mL) via oral gavage during (days 0–4) and two days (days 5, 6) after FOLFOX treatment (Figure 1A). During the experiment, FMT did not affect body weight in the saline and the FOLFOX groups (Figure 1B).

The size of the subcutaneously injected tumors was recorded. Though FMT (FMT50: 50 mg/mL or FMT150: 150 mg/mL) alone did not affect tumor growth (Figure 1C), in FOLFOX-challenged colon cancer-bearing mice, the tumor growth was significantly prevented by FOLFOX, as compared to that observed in controls. At the end of the study (day 6), the tumor size was significantly decreased in the FOLFOX group as compared to that of the control group (172.31% ± 49.70% vs. 355.4% ± 33.66%; *p* = 0.005). Furthermore, FMT did not impact the anti-tumorigenic activities of FOLFOX as compared to its effects on the FOLFOX group (day 6; FMT50 + FOLFOX vs. FOLFOX, *p* = 0.80; FMT150 + FOLFOX vs. FOLFOX, *p* = 0.94).

Diarrhea scores for all groups were recorded daily, and the results were compared. In saline-treated groups (subjected or not subjected to FMT), no significant diarrhea was detected, and their total scores were < 1 throughout the experiment. However, after receiving FOLFOX injections, the mice experienced the most severe diarrhea on day 5 (one day after the completion of FOLFOX treatment) (2.9 ± 0.1 vs. 0.2 ± 0.1, *p* < 0.005; Figure 1D). On day 6, the severity of diarrhea was clearly attenuated in mice treated with FMT50 and FMT150 in the FOLFOX groups (2.0 ± 0.15 and 2.1 ± 0.1 vs. 2.8 ± 0.13, *p* < 0.005; Figure 1D).

The survival was monitored for 30 days with Kaplan–Meier survival analysis. As shown in Figure 1E, we observed no mortality in all groups on day 6 before euthanasia; however, FOLFOX administration reduced animal survival in experimental animals monitored for 30 days, and FMT50 and FMT150 administration statistically improved survival in the FOLFOX group, as shown by the results of the log-rank test (*p* < 0.005). In addition, no mortality was observed up to day 30 of the experimental study in groups in which FMT50 and FMT150 alone were administered and the saline group.

With regard to the safety of FMT, we examined mesenteric lymph node, liver, and spleen tissue samples and blood and calculated the bacterial colonies for performing a translocation assay (Table 2). Mesenteric lymph node, liver, and spleen samples showed evidence of translocation more frequently in the FMT150 group than in the FMT50 group as compared to those of the saline group. The FOLFOX group showed a more frequent translocation of bacteria to the mesenteric lymph node, liver, and spleen as compared to that observed for the saline group. The administration of FMT50 and FMT150 tended to reduce bacterial translocation in the FOLFOX group. Furthermore, the total translocation ratio was significantly higher in FMT150, FOLFOX, and FOLFOX+FM150 groups—but not in the FMT50 or the FOLFOX+FMT50 groups—and was higher than that for the saline group (*p* < 0.05). At necropsy, no bacteria were detected to the bloodstream in any of the groups (Table 2).

### 2.2. The Effects of FMT on FOLFOX-Induced Intestinal Mucosal Damage in Colorectal Carcinoma-Implanted Mice

As expected, FOLFOX induced substantial histological changes in the intestinal mucosal layer at day 6 (Figure 2A), including flattening of the epithelium, shortening of the villi, infiltration of inflammatory cells into the lamina propria, and thickening of crypts; these are hallmarks of intestinal mucositis.

In the FOLFOX group, the intestinal villus height was significantly decreased in the jejunum as compared to that in the control groups (Figure 2B). FMT (FMT50: 50 mg/mL or FMT150: 150 mg/mL) significantly abrogated this effect in FOLFOX-injected mice and caused a significant lengthening of the villi as compared to that of the FOLFOX group (*p* < 0.001; Figure 2B). In the FOLFOX group, the intestinal crypt depth was significantly increased as compared to that of the control group (*p* < 0.001; Figure 2C), while this increase in crypt depth was significantly reversed with FMT50 and FMT150 administration (*p* < 0.001; Figure 2C). Villus height-to-crypt depth ratios were markedly decreased after FOLFOX administration in jejunal sections as compared to the ratio in the control groups. FMT50 and FMT150 administration could also abrogate these effects (*p* < 0.001; Figure 2D). No marked effects were observed on intestinal villus height, crypt depth, and their ratio after the administration of FMT50 alone (Figure 2A–D). Thus, an oral FMT consisting of a 50 mg/mL dose proved safe and effective for reducing FOLFOX-induced intestinal mucosal injury in colorectal carcinoma-implanted mice; this was determined after measuring villus height (344.16 ± 4.24 vs. 240.66 ± 4.04 μm, *p* < 0.001), crypt depth (82.73 ± 0.87 vs. 112.61 ± 1.49 μm, *p* < 0.001), and villus height-to-crypt depth ratio (4.07 ± 0.1 vs. 2.12 ± 0.1, *p* < 0.001).

### 2.3. The Effects of FMT on the Epithelial Barrier of Villi in Colorectal Cancer-Bearing Mice after FOLFOX Treatment

To confirm the association with intestinal barrier dysfunction, we found that, in comparison to the control group, periodic acid–schiff (PAS)/alcian blue (AB) staining showed a significant decrease in mucin-filled goblet cell number in the intestinal villus after FOLFOX was administered (3.45 ± 0.23 vs. 18.20 ± 0.98 cells/crypt, *p* < 0.001, Figure 3A,B). At a dose of 50 mg/mL, FMT significantly reduced goblet cell damage in the FMT50/FOLFOX group (13.98 ± 0.49 vs. 3.45 ± 0.23 cells/crypt, *p* < 0.001, Figure 3A,B), which suggested that FMT prevented villus damage and significantly affected goblet cell differentiation and mucus barrier functioning in the intestine.

Furthermore, we performed an immunohistochemical analysis of the intracellular tight junction protein zonula occludens-1 (ZO-1) in mouse small intestinal tissue. We found that the level of ZO-1 staining was significantly decreased in the intestinal villus after FOLFOX administration (49.87 ± 1.56 vs. 84.71 ± 5.1 mean intensity, *p* < 0.001, Figure 3A,C) as compared to that in the control group. A dose of 50 mg/mL FMT significantly counteracted this effect in the FMT50/FOLFOX group (78.39 ± 2.28 vs. 49.87 ± 1.56 mean intensity, *p* < 0.001, Figure 3A,C), which suggests that FMT prevented damage to the villi and significantly affected ZO-1 protein expression and barrier functioning in the intestine.

### 2.4. The Effects of FMT on Proliferation, Regeneration, and Apoptosis of Crypts in Colorectal Cancer-Bearing Mice after FOLFOX Treatment

To study proliferation in intestinal crypts, the crypt Ki67 expression level in jejunal segments (Figure 3A,C) was determined. In comparison to the saline group, the administration of FMT50 alone had no effect on the Ki67 level. Two days after FOLFOX administration, the number of Ki67-positive cells was significantly increased in the FOLFOX group as compared to that of the saline group (25.86 ± 1.48 vs. 9.06 ± 1.15 cells/crypt, *p* < 0.001; Figure 3A,D). FMT (FMT50: 50 mg/mL) administration significantly affected proliferative activity after FOLFOX was injected in the FMT50/FOLFOX group (3.91 ± 0.56 vs. 25.86 ± 1.48 cells/crypt, *p* < 0.001; Figure 3A,D).

An antibody against CD44 was used to stain cells in crypts to evaluate whether the crypts in jejunal segments were repopulated by intestinal progenitor cells. In comparison to the saline group, the administration of FMT50 alone had no effect on the level of CD44. The number of CD44-positive cells at the bottom of the crypts increased significantly after FOLFOX treatment (56.72 ± 3.12 vs. 5.4 ± 0.81 cells/crypt, *p* < 0.001; Figure 3A,E). FMT (FMT50: 50 mg/mL) administration significantly affected the regenerative activity after FOLFOX was administered into the FMT50/FOLFOX group (5.36 ± 0.78 vs. 56.72 ± 3.12 cells/crypt, *p* < 0.001; Figure 3A,E).

To determine the number of apoptotic cells in intestinal tissues, the terminal deoxynucleotidyl transferase dUTP nick-end labeling (TUNEL) assay was performed. FOLFOX administration caused a marked increase in TUNEL-positive apoptotic cells in intestinal crypts. The number of apoptotic cells in the FOLFOX-treated group was about six-fold that of the saline group (31.52 ± 4.49 vs. 4.89 ± 0.7 cells/crypt, *p* < 0.0001, Figure 3A,F). FMT50 significantly reduced the FOLFOX-induced increase in the number of apoptotic cells (5.93 ± 0.36 vs. 31.52 ± 4.49 cells/crypt, *p* < 0.0001, Figure 3A,F). FMT accelerated the disappearance of TUNEL-positive cells in the FMT50/FOLFOX group across the crypts of jejunal segments.

There were many p65-reactive cells in the crypts of FOLFOX-treated intestine in jejunal segments (32.94 ± 2.2 vs. 2.24 ± 0.21 cells/crypt, *p* < 0.001, Figure 3A,G) as compared to the saline group. FMT50 significantly reduced the FOLFOX-induced increase in the number of p65-positive cells in the intestine (10.55 ± 1.27 vs. 32.94 ± 2.2 cells/crypt, *p* < 0.001, Figure 3A,G). These findings indicated that FOLFOX induced NF-κB expression, while FMT inhibited FOLFOX-induced NF-κB expression across the crypts of jejunal segments.

### 2.5. The Effects of FMT on mRNA Expression Levels of Tight Junction Proteins and Toll-Like Receptors in the Jejunum of Colorectal Cancer-Bearing Mice Challenged with FOLFOX

After euthanasia, the effects of FMT (FMT50: 50 mg/mL) administration on mRNA expression of ZO-1, occludin, claudin-2, and junctional adhesion molecule-A (JAM-A) in the jejunum of mice treated with FOLFOX were determined. FMT50 treatment alone did not markedly affect the mRNA expression of ZO-1, occludin, claudin-2, and JAM-A (Figure 4A–D). *ZO-1* mRNA expression was markedly downregulated in the jejunal tissues of the FOLFOX-challenged group (0.12 ± 0.06 vs. 1.00 ± 0.09, *p* < 0.001; Figure 4A). FMT50 treatment significantly upregulated FOLFOX-induced ZO-1 mRNA suppression in jejunal tissues (0.6 ± 0.11 vs. 0.12 ± 0.06, *p* = 0.003; Figure 4A). Claudin-2 mRNA expression was markedly upregulated in the jejunal tissues of the FOLFOX-challenged group (3.02 ± 1.43 vs. 1.00 ± 0.21, *p* = 0.005; Figure 4C). FMT50 treatment significantly downregulated FOLFOX-induced Claudin-2 mRNA expression in jejunal tissues (0.22 ± 0.05 vs. 3.02 ± 1.43, *p* = 0.001; Figure 4C).

Previous studies have shown that intestinal epithelial cells express several TLRs, including TLR1, TLR2, TLR4, TLR5, and TLR6, in vitro and in vivo [25,26]. Figure 5A–E show the different expression patterns of TLR1, TLR2, TLR4, TLR5, and TLR6 mRNA in the jejunum of mice. FOLFOX treatment significantly increased the mRNA expression of TLR1 (101.00 ± 15.60 vs. 1.00 ± 0.40, *p* < 0.001), TLR2 (96.85 ± 17.10 vs. 1.00 ± 0.26, *p* < 0.001), TLR4 (110.96 ± 23.83 vs. 1.00 ± 0.32, *p* < 0.001), TLR5 (81.49 ± 21.47 vs. 1.00 ± 0.21, *p* < 0.001), and TLR6 (171.47 ± 27.09 vs. 1.00 ± 0.35, *p* < 0.001) in the jejunum of mice. The administration of FMT50 significantly reduced the expression levels of TLR1 (0.26 ± 0.08 vs. 101.00 ± 15.60, *p* < 0.001), TLR2 (0.59 ± 0.33 vs. 96.85 ± 17.10, *p* < 0.001), TLR4 (0.30 ± 0.04 vs. 110.96 ± 23.83, *p* < 0.001), TLR5 (0.79 ± 0.28 vs. 81.49 ± 21.47, *p* < 0.001), and TLR6 (0.13 ± 0.35 vs. 171.47 ± 27.09, *p* < 0.001) as compared to the levels in the FOLFOX group. However, no significant difference was detected between saline and FMT groups. Furthermore, FOLFOX treatment significantly increased the mRNA expression of MyD88 (3.88 ± 0.55 vs. 1.00 ± 0.24, *p* < 0.005, Figure 5F) in the jejunum of mice. The administration of FMT50 significantly reduced the expression levels of MyD88 (0.52 ±0.1 vs. 3.88 ± 0.55, *p* < 0.001, Figure 5F) as compared to those of the FOLFOX group.

### 2.6. The Effects of FMT on the Levels of Serum Inflammatory Cytokines in Colorectal Cancer-Bearing Mice Challenged with FOLFOX

After euthanasia, the effects of FMT treatment on serum levels of cytokines interleukin-1β (IL-1β), IL-6, tumor necrosis factor-α (TNF-α), and IL-10 in FOLFOX-treated mice were determined, the results of which are shown in Figure 6A–D. IL-1β and IL-6 levels were markedly increased (12.64 ± 3.27 vs. 5.50 ± 0.97, *p* = 0.03 and 24.73 ± 5.15 vs. 7.75 ± 0.67, *p* < 0.001, respectively; Figure 6A,B), while TNF-α, and IL-10 levels also showed a trend of increase in the FOLFOX-challenged group (Figure 6C,D). FMT treatment significantly suppressed IL-6 levels in the FOLFOX group (14.30 ± 2.26 vs. 24.73 ± 5.15, *p* < 0.05; Figure 6B).

### 2.7. The Effects of FMT on the Fecal Gut Microbiota Composition of FOLFOX-Challenged, Colorectal Cancer-Bearing Mice

The composition of gut microbiota in stool was evaluated by next-generation sequencing. Taxonomic analysis at the phylum level revealed that FOLFOX changed the composition of the gut microbiota. FMT altered this composition in the FOLFOX-challenged group as compared to that in the saline group (Figure 7A,B). The heatmap at the phylum levels revealed saline and FMT+FOLFOX groups clustered together, resembling a more distant FMT group. The FOLFOX group was the most different group (Figure 7B). Taxonomic analysis at the genus level also revealed that FOLFOX altered the composition of the gut microbiota. FMT altered this composition in the FOLFOX-challenged group as compared to that in the saline group (Figure 7C,D). The heatmap at the genus level revealed the saline and the FMT groups clustered together, resembling more distant FOLFOX and FMT+FOLFOX groups (Figure 7D). Bacteroidetes (B) and Firmicutes (F) were identified as the major phyla present in all groups. FOLFOX changed the abundance of the Bacteroidetes and the Firmicutes as compared to that in the saline group (Figure 7B). FMT altered this abundance in the FOLFOX-challenged group (Figure 7B). FOLFOX markedly increased relative abundance of the F/B ratio as compared to that of the saline control. These effects were abrogated by FMT50 administration in FOLFOX-injected mice (0.3 ± 0.04 vs. 0.73 ± 0.05 F/B ratio, *p* < 0.01; Figure 7E). The alpha diversity indexes, including the Shannon entropy (community diversity), were determined, and no significant differences were observed in the microbiota of saline, FMT, FOLFOX, and FOLFOX+FMT groups (Figure 7F). Beta diversity was analyzed using the Jaccard and Bray–Curtis distance matrix analysis method, as illustrated in Table 3, and it showed that there were significant differences in the beta diversity of the microbiota of saline, FMT, FOLFOX, and FOLFOX+FMT groups [*p* < 0.05; false discovery rate (FDR) corrected *p* < 0.05]; FMT and FOLFOX treatment alone affected the composition of fecal gut microbiota. Furthermore, the oral administration of FMT specifically altered FOLFOX-affected gut microbiota in colorectal carcinoma-implanted mice (*p* < 0.01; FDR corrected *p* < 0.05). Because a Jaccard analysis only accounts for the presence/absence of operational taxonomic units (OTUs), and a Bray–Curtis analysis is more sensitive to changes in OTU abundance, this supports the fact that the compositional changes in the fecal microbiota of FMT in the FOLFOX+FMT group were likely influenced by both a loss or a gain of species and changes in the level of abundance of present species.

## 3. Discussion

This is the first study to our knowledge to report on the potential and the safety of use of FMT for suppressing FOLFOX-induced mucositis in a colorectal cancer mouse model in vivo. FOLFOX administration significantly prevented tumor growth in colon cancer-bearing mice. FMT ameliorated FOLFOX-induced severe diarrhea, bacterial translocation, and intestinal mucosal injury and improved long-term survival associated with FOLFOX administration in CT26 colorectal cancer-bearing mice. Furthermore, FMT reduced FOLFOX-induced intestinal mucosal inflammation and barrier integrity disruption. This was characterized by immunohistological changes and expression of tight junction proteins. FOLFOX-induced intestinal mucosal apoptosis, changes in fecal gut microbiota, and TLR expression were also ameliorated by FMT. Therefore, these results indicate that FMT may have clinical potential for the management of chemotherapy-induced intestinal dysbiosis and toxicity.

The transplantation of fecal microbiota was performed in order to manipulate the gut microbiota in various inflammatory disease models in vivo [2,11,17,27,28,29]; however, few studies have investigated the effect of FMT on gastrointestinal mucositis resulting from combined 5-FU and oxaliplatin chemotherapy in animal models [2,11,17,27,28,29]. The evidence supporting the use of FMT for achieving improved chemotherapy toxicities is inconclusive, and studies investigating this topic are ongoing [2]. Li et al. revealed that fecal transplantation from healthy mice might alleviate weight loss, colon shortening, and intestinal mucositis induced by 5-FU in healthy BALB/c mice [30]. Our previous study reported that oral probiotic *Lcr35* prevented FOLFOX-induced intestinal mucositis in colorectal cancer-bearing mice [24]. The colorectal cancer-bearing mice model mimicked side effects of chemotherapy-associated gastrointestinal toxicity and diarrhea. Here, we showed that oral FMT significantly attenuated diarrhea and improved diarrhea scores in the FOLFOX groups. Furthermore, the histological analysis of the intestinal mucosal injury caused by FOLFOX in the mouse model was prevented by FMT. Our study revealed that FMT may ameliorate the severity of mucositis induced by chemotherapy.

In mucosal barrier integrity, primarily the mucus layer, the tight junctions, and the intestinal epithelial cells, dysfunction plays an important role in the pathogenesis of chemotherapy-induced gut toxicity and associated diarrhea [31]. This mucus layer is composed of mucins, which are diverse, complex glycoproteins secreted by the goblet cells. Tight junctions seal the paracellular space of the intestinal epithelium and maintain structure and function of the intestinal mucosal barrier, which is disrupted under inflammation conditions [32,33]. Tight junctions are composed of core proteins such as the transmembrane proteins, including claudins (CLDNs), occludin, and junctional adhesion molecules (JAMs), and these molecules bind directly to periplasmic (intracellular) scaffolding proteins such as zonula occludens (ZOs). ZOs are the most important components for the construction of a constitutive barrier of epithelial cells, and they regulate the permeability of the barrier by tightly sealing the cell–cell junctions [34]. Several potential markers of the intestinal barrier, the ileum, or the colon can be assessed in mice, including mucus thickness, mRNA expression, and levels of Muc-2 and junction molecules (JAM-A, ZO-1, occludin, claudin-1, -2, and -5) [35]. Chemotherapeutic drugs could increase intestinal epithelial barrier permeability by reducing tight junction expression in vivo. Song et al. investigated the 5-fluorouracil-induced changes in intestinal integrity biomarkers in BALB/C mice. They showed that the expression levels of occludin and claudin-1 were significantly reduced, while that of ZO-1 was unchanged in the small intestines of mice after injecting 5-FU once as compared to those of the control group [36]. Li et al. revealed that, while levels of the tight junction protein occludin were reduced, levels of ZO-1 and junctional adhesion molecule-A (JAM-A) were increased in colonic tissues of mice by injecting 5-FU once daily for three days [30]. The effects of chemotherapeutic drugs such as 5-FU on the intestinal epithelial barrier via the expression of tight junction proteins are reported inconsistently [30,31,36,37] This might be attributable to differences in regimens, experimental protocols, or animal models used. Our study also revealed that FMT prevented FOLFOX-induced villus damage by significantly affecting the differentiation of mucin-filled goblet cells in the intestine. Thus, FMT prevented FOLFOX-induced dysfunction in mucosal barrier integrity and affected goblet cell differentiation and intercellular scaffolding tight junction expression of proteins such as ZO-1. This suggests that FMT could reduce FOLFOX-induced mucosal barrier damage in colorectal tumor-bearing mice.

Current studies on the safety of FMT in patients or animal models with tumors and in the context of treatment with anti-neoplastic agents are limited [16,17,18,19]. Cancer patients not only have a high mortality because of individual episodes of sepsis but suffer disproportionately from sepsis as compared to the general population [38]. The increased risk of sepsis in cancer patients has been attributed to myelosuppression, mucositis, and the use of central venous catheters [39]. Gut pathogens are an important cause of sepsis in cancer patients. Damage to the gut epithelium after chemotherapy, cancer itself, and bacterial overgrowth contribute to bacterial translocation, making those receiving chemotherapy particularly vulnerable to bloodstream infections caused by enteric bacteria [40]. Because feces act as mediators between the donor and the recipient, FMT has the potential to transmit occult infections, even when donor screening is performed stringently [16,17]. Analyses made during a short-term follow-up period showed that it was safe to perform FMT; however, recent reports show that adverse events occurring after FMT were reportedly associated with the mortality rate for *Clostridium difficile* infections, including septic shock with decompensated toxic megacolon and fatal aspiration pneumonia [16,17]. Therefore, the safety of the FMT process continues to limit its use in immunocompromised and cancer patients treated with anti-neoplastic agents, and its long-term side effects, if any, remain unknown [16,17,19]. However, FMT might reportedly be able to correct dysbiosis and prevent bacterial translocation and sepsis and improve survival. For instance, Li et at showed that, in a mouse model, FMT could not only reverse bacterial translocation but could also improve the survival period associated with experimental necrotizing enterocolitis [28]. FMT caused the reversal of survival in a BALB/c mouse model of ulcerative colitis [27]. FMT attenuated radiation-induced toxicity, which eventually led to death in a mouse model [29]. Our studies showed that, in a colorectal cancer-bearing mouse model, at necropsy, none of the bacteria were detected to the bloodstream in all experimental groups; however, bacterial translocation to mesenteric lymph nodes, liver, and spleen was more frequent in the FOLFOX-treated group than in the saline group. FMT tended to reduce bacterial translocation in the FOLFOX-treated group. In addition, there is a potential risk of treatment-related mortality in cancer patients receiving FOLFOX [41]. In our chemotherapy-related mouse model with gastrointestinal toxicity, we also observed that FOLFOX administration reduced mouse survival. FMT increased the survival rate of FOLFOX-treated mice in the long-term experimental study, suggesting that FMT might safely provide protection against chemotherapy-induced toxicity and mortality in this colorectal cancer-bearing mouse model treated with anti-neoplastic agents.

TLRs act as sensors for microbial infection and are critical for the initiation of inflammation and immune defense responses [42]. TLRs recognize ligand from fungi, viruses, and bacteria. Bacterial ligands are not only unique to pathogens but are found in all bacteria and are produced by symbiotic microorganisms [43]. During chemotherapy, chemotherapeutic agents can reduce the number and the diversity of microbiota and cause dysbiosis [6]. TLRs can sense dysbiosis and danger signals from dead or injured host cells. Upon activation, each TLR differentially recruits members of a set of TIR domain-containing adaptors, such as MyD88, TRIF, TIRAP/MAL, or TRAM. Collectively, depending on adaptor usage, TLR signaling is largely divided into two pathways, i.e., the MyD88-dependent and the TRIF-dependent pathways. The subsequent transcriptional activation of TLR target genes encoding pro-inflammatory and anti-inflammatory cytokines, chemokines, effector molecules, and type I interferons initiates the activation of antigen-specific and non-specific adaptive immune responses [44]. MyD88 is utilized by all TLRs and activates NF-κB and MAPKs for the subsequent induction of inflammatory cytokine genes, such as granulocyte colony-stimulating factor, interleukin-1β, IL-6, and TNF-α. Thus, MyD88 and downstream NF-κB are the key signaling molecules involved in the MyD88-dependent TLR signaling pathway [45,46,47,48]. Previous studies have shown that intestinal epithelial cells expressed several TLRs, including TLR1, TLR2, TLR4, TLR5, and TLR6 in vitro and in vivo, and that these TLRs were expressed on the cell surface and mainly recognized microbial membrane components, such as lipids, lipoproteins, and proteins [25,26,46,49]. TLR4 recognizes bacterial lipopolysaccharide (LPS). TLR5 recognizes bacterial flagellin [50]. The heterodimers formed by combinations of TLR1, TLR2, and TLR6 recognize a wide variety of pathogen-associated molecular patterns, including lipoproteins, peptidoglycans, lipoteichoic acids, zymosan, mannan, and tGPI-mucin [25,26,46,49]. In our study, FOLFOX treatment might have disturbed the original balance of gut microbiota and significantly upregulated the mRNA expression of TLRs (TLR1, TLR2, TLR4, TLR5, and TLR6) and key signal molecule MyD88 in the intestine of mice. FMT significantly suppressed the expression of TLRs and MyD88 in the FOLFOX group. Thus, our data suggest that FMT can suppress the effects of FOLFOX and induce the MyD88-dependent TLR signaling pathway; this indicates that intestinal microbiota of recipient mice in the FOLFOX group and homeostasis were reestablished.

The mechanisms by which the clinical benefits of FMT are achieved, including in inflammatory disease patients, are not completely understood. Several potential mechanisms might primarily include restoration of the intestinal microenvironment, direct interaction of donor gut microbiota with the host physiology, gut mucosal barrier, and immune system [13,14]. NF-κB is a transcription factor that regulates the expression of numerous genes that are critical for survival and control inflammation, cell growth, apoptosis, and the cell cycle [9]. NF-κB plays a central role in the pathobiology of the five-phase model for the development of chemotherapy-induced intestinal mucositis and acts as a “gatekeeper” for various pathways [9,51]. Its activation is induced by anti-neoplastic agents, such as 5-FU, which are thought to elicit inflammatory and apoptotic responses in the intestine [9,51]. NF-κB activation triggers the activation of multiple signaling pathways for the synthesis of different pro-inflammatory cytokines involved in various stages of mucositis, including TNF, IL-1β, and IL-6 [9]. NF-κB is also the key signaling molecule of the TLR signaling pathway involved in the induction of inflammatory cytokine genes (such as IL-1β, IL-6, and TNF-α) [45,46,47,48]. In addition, a number of studies have demonstrated a change in tight junction permeability attributable to TNF-α, IL-1β, and IL-6, and these changes in tight junctions were involved in the functioning of the NF-κB signaling pathway [52,53]. In this study, immunohistological analysis indicated that FMT decreased FOLFOX-induced NF-κB activity in the intestine and reduced FOLFOX-induced apoptosis. We also showed that serum levels of IL-1β and IL-6 were significant, and that the TNF-α level tended to be increased in the FOLFOX group, while FMT abrogated the effect of IL-6. These findings suggest that the inhibition of NF-κB activity by FMT might result in the suppression of inflammation and the sequential amelioration of FOLFOX-induced mucosal barrier damage and apoptosis in the intestine. Accordingly, our data suggest that, by modulating the composition of the gut flora, FMT altered FOLFOX-induced changes in the gut microbiota and influenced the pathogenesis of mucositis via the gut microbiota-TLR-NF-κB signaling pathway in colorectal carcinoma-implanted mice.

Gut microbiota homeostasis has been manipulated via FMT for treating various local and systemic inflammatory diseases linked to gut microbiota models and clinical trials, such as inflammatory bowel disease, radiotherapy-induced diarrhea, obesity, and diabetes [2,10,11,54,55]. Chemotherapy is associated with changes in microbial diversity [4,5,6,54]. This change in microbial diversity coincides with the development of severe chemotherapy-induced mucositis; commensal intestinal bacteria could be significantly involved in the pathogenesis of mucositis [54]. Some studies demonstrated that the gut microbiota was actively involved in the pathogenesis of 5-FU induced intestinal mucositis, which suggests that 5-FU induced intestinal mucositis could be potentially attenuated by disturbing the homeostasis of gut microbiota [2,24,30,56]. Accordingly, the disturbance in the homeostasis of gut microbiota could have therapeutic effects against chemotherapy-induced mucositis. Here, FOLFOX changed the composition of gut microbiota, and the oral FMT altered this compositional change. Further taxonomic analyses at the phylum level indicated that FOLFOX significantly increased the abundance ratio of Firmicutes/Bacteroidetes (F/B). These changes were restored by FMT. Bacteroidetes and Firmicutes are the predominant phyla in humans and mice [6,57]. The balance between these two phyla (F/B ratio) appears to be critical for the regulation of radiotherapy or chemotherapy-related mucositis [6,24,58]. Le Bastard et al. showed that 5-FU-induced gut dysbiosis, richness, and diversity were restored by FMT in a healthy mouse model [59]. In tumor-bearing mice, our study revealed that the alpha diversity indexes, including the Shannon (community diversity) index, were determined, and no significant differences were found with regard to the microbiota of saline, FMT, FOLFOX, and FOLFOX+FMT groups. However, beta diversity analysis using the Jaccard and the Bray–Curtis distance matrix analysis method showed FMT and FOLFOX treatment alone affected the composition of the fecal gut microbiota. Because a Jaccard analysis accounts merely for the presence/absence of OTUs, and a Bray–Curtis analysis is more sensitive to changes in OTU abundance [60], the compositional changes in the fecal microbiota used for FMT in the FOLFOX+FMT group were probably influenced by the exclusion or the inclusion of certain species and changes in the abundance of present species.

Our study also has some limitations. We did not work in anaerobic conditions for preparation of frozen fecal material. The sample size of mice was small, and mice were co-housed, not single-housed, for this study. During co-housing, animals may feed on feces (also known as coprophagy) or ingest feces by self-grooming [61]. It is a risk of transmission of gut microbiota between littermates [61]. The study may need to be repeated and validated in different groups of mice single-housed in the future before clinical trials of FMT. We did not conduct an analysis of time-course and dose-dependent effects of FMT; it would have provided more information regarding their role in the pathogenesis of chemotherapy-induced mucositis. We did not conduct further in vivo studies of gene knockout or knockdown or in vitro studies, which would have elucidated the anti-mucositis mechanism. Our microbiota studies were small in size; only fecal samples, not colon content, were collected, and we did not identify the specific microorganisms associated with chemotherapy injuries and the functional roles of these microorganisms in the gastrointestinal tract. Clinical studies also need to be performed to demonstrate the beneficial effects of FMT and elucidate safety and effective regimens for the management of chemotherapy-induced mucositis.

## 4. Materials and Methods

### 4.1. Animals

Six to eight week old male BALB/c mice with a weight of 22–24 g were purchased from the National Laboratory Animal Center (Taipei, Taiwan) and co-housed in a rodent facility at a temperature of 22 ± 1 °C, humidity of 55% ± 10%, and in a 12-h light–dark cycle [24,62]. All the animal studies were approved by the Institutional Animal Care and Use Committee (IACUC) of MacKay Memorial Hospital (Taiwan) (IACUC Number: MMH-A-S-106-30, approved on 1 May 2017 and performed in accordance with institutional ethical guidelines. Mice were provided with water and laboratory chow (laboratory-autoclavable rodent diet 5010) ad libitum.

### 4.2. Cell Culture

The CT26 colon cancer cell lines, *N*-nitroso-*N*-methyl urethane-induced mouse colon carcinoma cells of BALB/c origin [63], were purchased from the American Type Culture Collection (ATCC, Manassas, VA, USA). Cells were cultured in a humidified incubator with 5% CO_2_ at 37 °C, grown in RPMI-1640 medium (Gibco, Grand Island, NY, USA) supplemented with 10% heat-inactivated fetal calf serum (Hyclone, Logan, UT, USA), passaged every 2–3 days with TEG solution (0.25% trypsin, 0.1% EDTA, and 0.05% glucose in Hanks’ balanced salt solution), and maintained in a state of exponential growth.

### 4.3. Tumor Implantation

The mice, randomly divided into control and experimental groups, were all subcutaneously implanted with CT26 cells (4 × 10^6^ cells) in the right gluteal region. When the diameter of the tumors was 0.5 cm, treatment was started (day 0); the mice were injected intraperitoneally (i.p.) with saline or a FOLFOX regimen for 5 days (days 0–4) (Figure 1A). For evaluating the effect of FMT, mice were orally administered saline or FMT suspension daily during and two days after FOLFOX administration for a total of 7 days (days 0–6) (Table 1 and Figure 1A).

### 4.4. Transplantation of Fecal Microbiota

Fecal samples were collected for performing FMT from 6–10 healthy, untreated, age-matched BALB/c donor mice (6–8 weeks of age) and were pooled and stored at −80 °C in a freezer. Stool was collected and homogenized in sterile normal saline (FMT50: 50 mg/mL, FMT150: 150 mg/mL) and centrifuged for 30 s at 3000 rpm at 4 °C to pellet the particulate matter. The supernatant was used for transplantation [64]. Each recipient mouse received 200 μL of the fecal slurry per day at an FMT-50 or an FMT-150 dose via oral gavage.

### 4.5. Chemotherapy Regimen

A combination chemotherapy regimen (FOLFOX) of 5-fluorouracil (5-FU, Sigma F6627), leucovorin (LV, Sigma F7878), and oxaliplatin (Sigma O9512) was administered intraperitoneally to induce mucositis and diarrhea. The drug-dosing scheme was based on that described in previous studies [24,65,66]. From days 0–4, the experimental animals received 5-FU and LV (30 and 10 mg/kg, i.p., respectively) once daily for five consecutive days. On the first day (day 0), the mice also received oxaliplatin (1 mg/kg, i.p.) 1 h after 5-FU/LV administration. Saline was injected i.p. in the control groups. 

### 4.6. Disease Severity Evaluation

Changes in body weight and tumor size as well as diarrhea severity and survival rate were recorded daily by a single observer to assess the disease severity. The change in mouse body weight was measured in grams (g) and expressed as a percentage (%), with regard to that at day 0. The largest (a) and the smallest (b) diameters of the tumor were measured using a caliper, and the tumor size was estimated according to the formula 0.5ab2 [67]. The Bowen′s score system was used to assess diarrhea severity [68] and to classify it into four grades based on the stool consistency as grades 0 (normal; normal stool or absent); 1 (mild diarrhea; slightly wet and soft stool); 2 (moderate diarrhea; wet and unformed stool); and 3 (severe diarrhea; watery stool). Two days after the administration of FOLFOX, mice were euthanized, and blood and tissue samples were harvested for biochemical and histological analyses (Figure 1A). Survival analysis was performed up to day 30 according to the method described by Kaplan–Meier using IBM SPSS software (version 21.0; SPSS Institute).

### 4.7. Safety of FMT: Translocation of Gut Microbiota

Specimens obtained from mesenteric lymph nodes (MLNs), spleen, liver, and blood were inoculated in MRS (Oxoid, Basingstoke, UK) broth for 7 days; the samples were homogenized. Then, 0.1 mL of these samples was seeded on MRS agar plates for 2 days, after which the bacterial colonies were counted to analyze the level of bacterial translocation [69]. The growth of bacteria was considered as evidence of the occurrence of bacterial translocation. The translocation ratio (TR) was expressed as a ratio of the total number of positive MLN, spleen, liver, and blood cultures to the total number of MLN, spleen, liver, and blood cultures in each group [70]. Bacteria were identified based on their colony characteristics and morphological features.

### 4.8. Histological Analyses

After euthanasia on day 6, intestinal tissues that were close to the duodenojejunal flexure of the proximal jejunum were removed. Each harvested and processed jejunal specimen was fixed in 10% buffered neutral formalin for 2 h, dehydrated in solutions with an increasing concentration of ethanol, cleared in xylene, and embedded in paraffin. Sections with a thickness of 4 μm were cut, mounted on glass slides, and stained with hematoxylin and eosin (H&E) [5]. Using a 20× magnification objective, the specimen images were scanned with a Tissue FAXS automatic scanning system, digitized, and analyzed using HistoQuest software (TissueGnostics, Vienna, Austria) [71].Villus height, crypt depth, and villus height-to-crypt depth ratio of each jejunal tissue section in the small intestines of each mouse were measured to determine if villi and crypts were whole and well-oriented.

### 4.9. Immunohistochemical Staining

Sections of jejunal tissue were dewaxed with xylene and gradually hydrated. Then, 10 mM sodium citrate (pH 6) or EDTA (pH 8) buffer was used at 98 °C for 15 min to conduct heat-induced antigen retrieval and quench the endogenous peroxidase activity using hydrogen peroxide for 10 min. The sections were incubated in protein blocking solution for 10 min, rinsed with TBST, and incubated with primary antibodies at the indicated dilutions at room temperature for 1 h or at 4 °C overnight. The antibodies used were as follows: anti-Ki-67 (ab16667; Abcam, 1:200 dilution); anti-CD44 (ab157107; Abcam, 1:2000 dilution); anti-NF-κB (ab28856; Abcam, 1:50 dilution); and anti-ZO-1 (ab157107, 1:100 dilution). Negative control sections were processed without primary antibodies. For detection, the Polink-2 plus Polymer horseradish peroxidase (HRP) Detection System (GBI Labs, Mukilteo) was used, and 3,3-diaminobenzidine was used for histological staining as a chromogen. The intestinal specimens were counterstained with hematoxylin. Apoptosis was assessed using the TUNEL (In Situ Cell Death Detection Kit, POD, Roche) to detect DNA breaks in cells. The PAS/AB stain was used for staining goblet cells in the intestinal sections (Alcian Blue-PAS Stain Kit; ScyTek Laboratories). Multiple intestinal tissue images (seven) that were observed to be stained with PAS/AB, ZO-1, Ki67, CD44, TUNEL, and NF-κB on the high-power fields (200×) of four groups were obtained, and the staining intensity was quantified by using the Tissue FAXS automatic scanning system and HistoQuest software (TissueGnostics, Vienna, Austria).

### 4.10. Inflammatory Cytokine Analysis

After mice were sacrificed, blood samples were immediately collected from the heart and centrifuged to obtain serum. The serum (50 l) levels of TNF-α, IL-1β, IL-6, and IL-10 were measured using Bio-Plex Cytokine 23-Plex (Bio-Rad, CA, USA) and quantified using an automated microsphere-based reader (BioPlex^®^ 200 system, Bio-Rad, CA, USA), according to the manufacturer′s instructions.

### 4.11. Relative Expression of mRNA Using Real-Time Quantitative (q)PCR

RNA extraction from the jejunal specimens was performed using the Trizol reagent (Invitrogen, Carlsbad, CA). Two micrograms of total RNA were used for cDNA synthesis with random hexamers. DNA detection and amplification via qPCR was performed by using an ABI 7500 Fast System v1.4.0 (Applied Biosystems) [5]. The mRNA levels were measured using real-time PCR with SYBR Green. Complementary DNA (cDNA) was generated by the reverse transcription of total RNA using the SuperScript III kit (Invitrogen). PCR cycles were carried out at 50 °C for 2 min, 95 °C for 10 min, and 40 cycles were carried out at 95 °C for 15 s and 60 °C for 1 min. Pairs of oligonucleotide primers specific to ZO-1, occludin, claudin-2, JAM-A, TLR1, TLR2, TLR4, TLR5, TLR6 and MyD88 are listed in Table 4. Gene expression was normalized to the GAPDH expression levels by using the following formula: ΔCt = (Ct of GAPDH − Ct of the gene).

### 4.12. Gut Microbiota Analysis

#### 4.12.1. Sample Collection and DNA Extraction

Fresh fecal samples were collected from mice and immediately stored at −80 °C. DNA from fecal material was extracted by using the QIAmp^®^ DNA stool mini kit (Qiagen, Germany). We stored the extracted DNA at −20 °C by dividing it into several tubes upon isolation before freezing following the manufacturer′s instructions without performing a repetitive freeze/thaw cycle.

#### 4.12.2. Sequencing and Analysis of 16S rRNA

The V3–V4 region of the 16S rRNA gene was amplified and sequenced using the Illumina Miseq platform, according to the manufacturer′s instructions. Specific primers (16S Amplicon PCR forward primer: 5′ TCGTCGGCAGCGTCAGATGTGTATAAGAGACAGCCTACGGGNGGCWGCAG; 16S Amplicon PCR reverse primer: 5′ GTCTCGTGGGCTCGGAGATGTGTATAAGAGACAGGACTACHVGGGTATCTAATCC) were used [72]. PCR amplification was carried out for each sample using 25 µL of the reaction mixture. Each reaction mixture contained 0.2 µM of each primer and 12.5 µL of the 2X KAPA HiFi HotStart ReadyMix (ROCHE, South Africa). PCR was performed by carrying out a step at 95 °C for 3 min, followed by 25 cycles at 95 °C for 30 sec, 55 °C for 30 sec, and 72 °C for 30 sec; then, an extension step was carried out at 72 °C for 5 min in a DNA thermocycler (Veriti 96 well Thermal Cycler, Applied Biosystems, Singapore). Then, we used AMPure XP beads (BECKMAN COULTER, USA) to purify the 16S V3 and V4 amplicons from the free primers and the primer dimer species. In each library, the stool PCR products were associated with dual indices and Illumina sequencing adapters using the Nextera XT Index Kit. This step used AMPure XP beads to purify products from free primers and primer dimer species. Quantification was carried out using a NanoDrop ND2000 spectrophotometer (Thermo Scientific, USA), Qubit dsDNA HS Assay Kit (Invitrogen, USA), and Labchip GX Touch 24 system (Applied Biosystems, USA). Then, the products were sequenced using the Miseq reagent kit v3 (600 cycles, 25M reads) (Illumina, cat. No. MS-102-300). The products were sequenced using Miseq (Illumina, USA). For further bioinformatic analysis, the sequences were produced by a sequencer according to the manufacturer′s instructions. CLC Genomics Workbench 12 software (CLC Bio, USA) was used along with the Greengenes 16S rRNA Taxonomy Database (gg_13_8) to conduct 16S rDNA analysis. “OTU clustering” function was used with de novo OTU clustering default parameter. “OTU clustering” function clusters similar reads based on 97% similarity. The minimal criteria of OTU is 10 as default. The OTU clusters are identified and used following analysis and data visualization. 

### 4.13. Statistical Analysis

Results are presented as the mean ± standard error of the mean (SEM). The incidence of bacterial translocation was statistically evaluated using the chi-square test. The log-rank (Mantel–Cox) test was used to determine differences between survival curves. Comparisons between two groups were made using the Student′s t-test. Comparisons between more than three groups were made using one-way ANOVA. Data were analyzed using IBM SPSS software (version 21.0; SPSS Institute). Taxonomy (i.e., phyla and OTUs) and α-diversity were analyzed by one-way Kruskal–Wallis H test followed by Mann–Whitney U post hoc test. The community structure (β-diversity) was analyzed by performing permutational multivariate analysis of variance (PERMANOVA) of ranked Bray–Curtis and Jaccard distances using the CLC Genomics Workbench 12 software (CLC Bio, USA). A PERMANOVA analysis for each pair of groups and the results of the test (pseudo-f statistic and p-value). False discovery rate (FDR) (Benjamini & Hochberg, 1995), which corrected for multiple testing, was also shown. The FDRs were calculated using R software [version 3.6.0 R package “stats”, function “p.adjust”, correction (“BH” or its alias “fdr”)]. The results were considered to be statistically significant if *p* < 0.05.

## 5. Conclusions

FMT could safely ameliorate inflammation, protect the epithelium by maintaining intestinal epithelial integrity, and reduce the severity of mucositis following FOLFOX treatment. The possible mechanisms may involve the gut microbiota TLR-MyD88-NF-κB signaling pathway in mice with implanted colorectal carcinoma cells. FMT may improve the survival rate of chemotherapy-treated mice. Different groups of mice should be analyzed before FMT can be recommended for clinical trials in cancer patients undergoing chemotherapy.

## Figures and Tables

**Figure 1 ijms-21-00386-f001:**
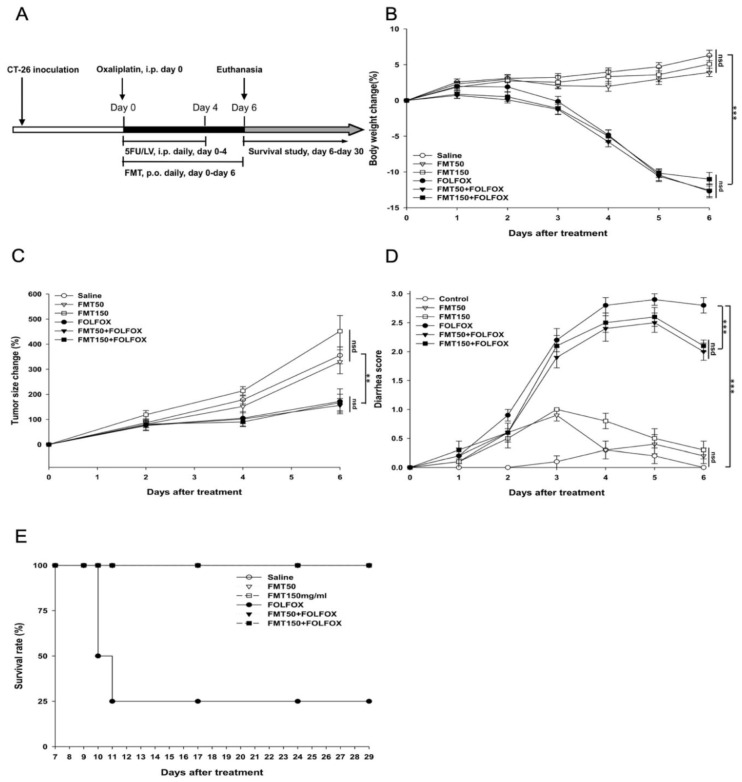
FOLFOX-challenged subcutaneously injected colorectal cancer mice with/without FMT (*n* = 10 for each group). (**A**) Protocols. (**B**) Percentage change in body weight; body weights are expressed as a percentage of the weight at day 0. (**C**) Percentage anti-tumor activity; tumor sizes are expressed as a percentage of the tumor size at day 0. (**D**) Diarrhea severity. (**E**) Survival study. Mice from each group with colorectal cancer were orally administered saline or FMT suspension at a dose of 200 µl/day (FMT50: 50 mg/mL or FMT150: 150 mg/mL) during and 2 days after intraperitoneal (i.p.) FOLFOX administration for a total of 7 days. Mice were euthanized 2 days after complete FOLFOX administration to evaluate the effects of FMT (*n* = 6 for each group), and each group of four mice was analyzed for the survival study. Details of the experimental procedures are given in the “Materials and Methods” section. nsd; no significant difference. *: *p*< 0.05, **: *p* < 0.01, ***: *p* < 0.001.

**Figure 2 ijms-21-00386-f002:**
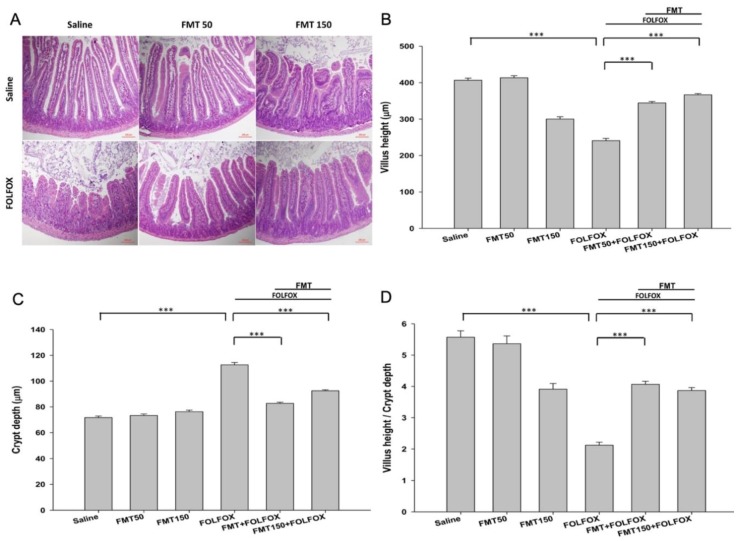
Histological changes in the jejunum of subcutaneously injected mice with colorectal carcinoma exhibiting FOLFOX-induced intestinal mucositis with/without FMT (*n* = 6 for each group). Segments of the jejunum were harvested for (**A**) hematoxylin and eosin staining (scale bar = 100 µm) and measuring (**B**) villus height, (**C**) crypt depth, and (**D**) villus height-to-crypt depth ratio per mouse. FMT50: 50 mg/mL. FMT150: 150 mg/mL. Values are presented as mean ± SEM. ***: *p* < 0.001.

**Figure 3 ijms-21-00386-f003:**
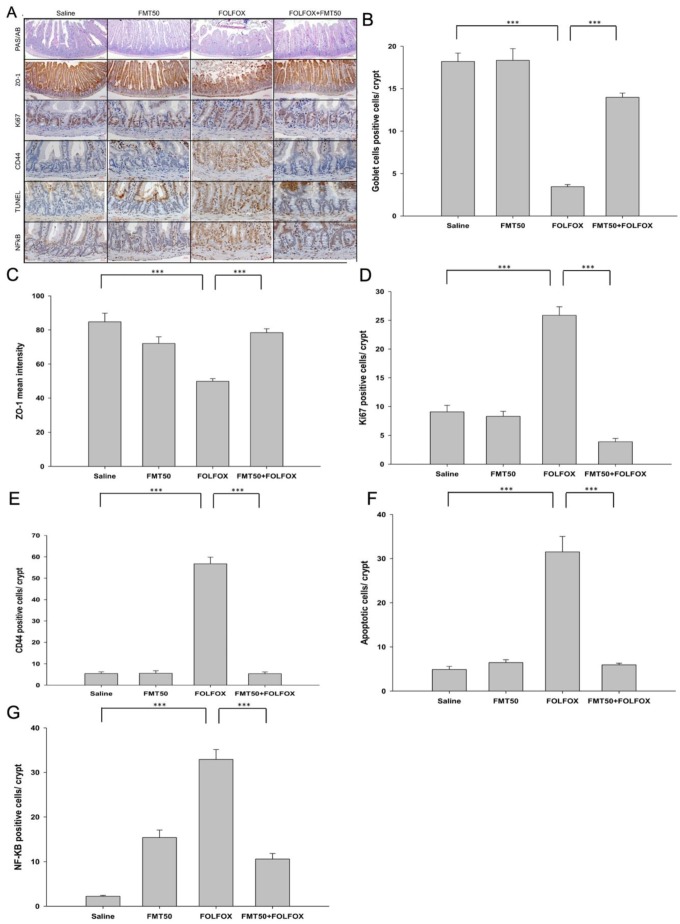
Activity of goblet cells, tight junctions, proliferation, regeneration, apoptosis, and NF-κB activity in subcutaneously injected mice with colorectal carcinoma presenting FOLFOX-induced intestinal damage with/without FMT (*n* = 6 for each group). (**A**) Immunohistochemical staining of jejunum sections was used to determine the effects of FMT on periodic acid–schiff (PAS) dark purple)/alcian blue (**A**,**B**) staining (goblet cells); zonula occludens-1 (ZO-1) (brown) to detect tight junction integrity; Ki67 (brown) expression to detect proliferative activity; CD44 (brown) staining to detect regeneration activity; terminal deoxynucleotidyl transferase dUTP nick-end labeling (TUNEL) (brown) to detect DNA breaks; and NF-κB activity (brown) in the intestine using antibodies. We quantified the staining intensity of (**B**) PAS/AB and (**C**) ZO-1 in the intestinal villus, (**D**) Ki67, (**E**) CD44, (**F**) TUNEL, and (**G**) NF-κB in the intestinal crypt in (**A**). Scale bar = 100 µm. FMT50: 50 mg/mL. Values are presented as the mean ± SEM. ***: *p* < 0.001.

**Figure 4 ijms-21-00386-f004:**
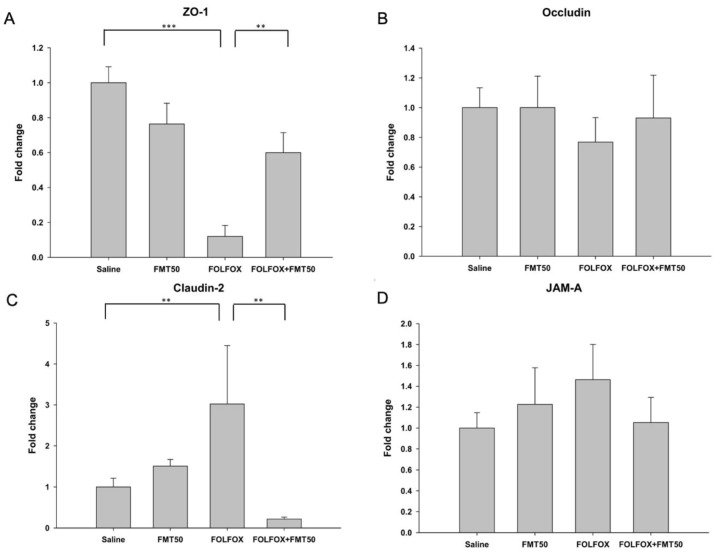
Tight junction mRNA expression levels in the jejunum of subcutaneously injected colorectal cancer mice exhibiting FOLFOX-induced intestinal damage with/without FMT (*n* = 6 for each group). Relative mRNA expression levels of the tight junction molecules (**A**) ZO-1, (**B**) occludin, (**C**) claudin-2, and (**D**) junctional adhesion molecule-A (JAM-A) were determined by qPCR in jejunum tissues. FMT50: 50 mg/mL. Values are presented as the mean ± SEM. **: *p* < 0.01, ***: *p* < 0.001.

**Figure 5 ijms-21-00386-f005:**
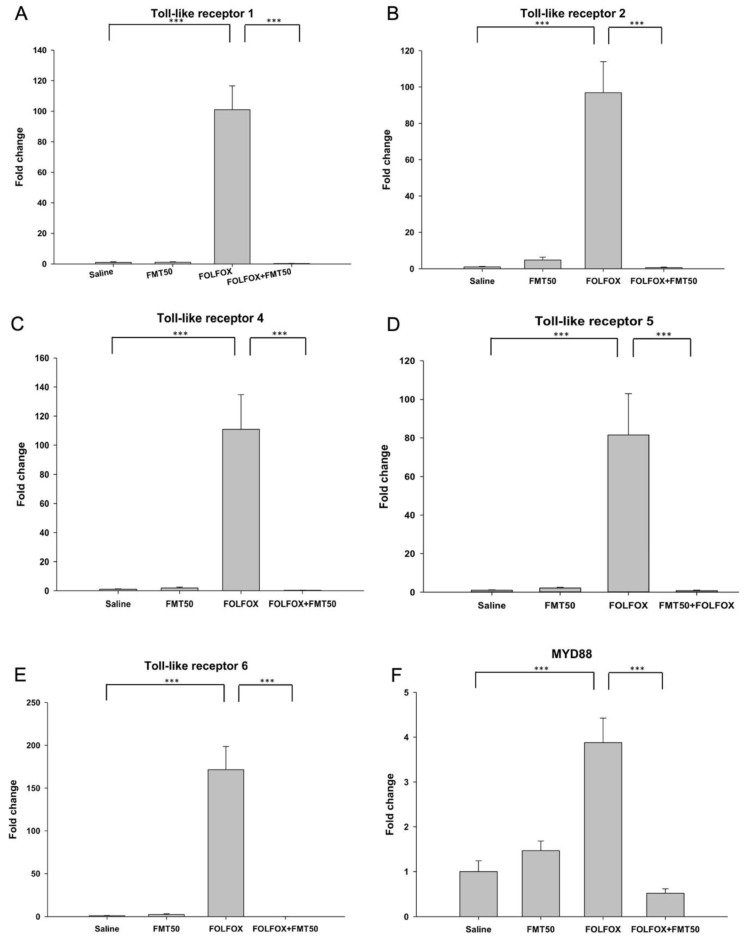
mRNA expression levels of toll-like receptors (TLRs) and MyD88 in the jejunum of subcutaneously injected colorectal cancer mice exhibiting FOLFOX-induced intestinal damage with/without FMT (*n* = 6 for each group). Relative mRNA expression levels of (**A**) TLR1, (**B**) TLR2, (**C**) TLR3, (**D**) TLR4, (**E**) TLR5, and (**F**) MyD88 were determined by qPCR in jejunum tissues. FMT50: 50 mg/mL. Values are presented as the mean ± SEM. ***: *p* < 0.001.

**Figure 6 ijms-21-00386-f006:**
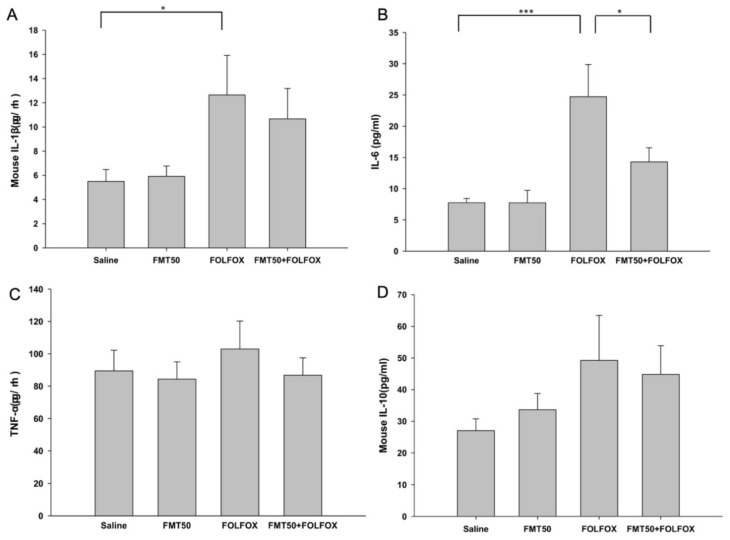
Serum protein levels of (**A**) interleukin-1β (Il-1β), (**B**) Il-6, (**C**) tumor necrosis factor-α (TNF-α), and (**D**) Il-10 in subcutaneously injected colorectal carcinoma mice exhibiting FOLFOX-induced intestinal damage with/without FMT (*n* = 6 for each group). FMT50: 50 mg/mL. Values are presented as the mean ± SEM. *: *p*< 0.05, ***: *p* < 0.001.

**Figure 7 ijms-21-00386-f007:**
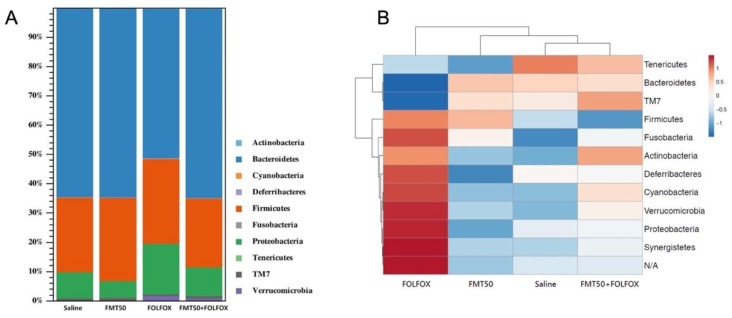
Changes in fecal gut microbiota of subcutaneously injected colorectal cancer mice presenting FOLFOX-induced intestinal damage with/without FMT (*n* = 5 for each group). The composition of gut microbiota was determined by the following: (**A**) bar charts, (**B**) heatmap of the relative abundance of gut microbiota at the phylum level, and (**C**) bar charts, (**D**) heatmap at the genus level, (**E**) abundance of Firmicutes-to-Bacteroidetes (**F**/**B**) ratio and (**F**) Tukey box plots of α-diversity. FMT50: 50 mg/mL. Values are presented as the mean ± SEM. **: *p* < 0.01.

**Table 1 ijms-21-00386-t001:** Colorectal cancer-bearing mice (*n* = 10 in each group) groups in which the effects of fecal microbiota transplant (FMT) on FOLFOX-induced intestinal mucositis were evaluated.

Groups	Regimen
1	Saline (untreated control)
2	FMT50
3	FMT150
4	FOLFOX
5	FOLFOX+FMT50
6	FOLFOX+FMT150

FMT50: 50 mg/mL, FMT150: 150 mg/mL.

**Table 2 ijms-21-00386-t002:** Analysis of translocation of bacteria to the mesenteric lymph node, blood, liver, and spleen of FOLFOX-treated mice subjected or not subjected to FMT on the 7th day (*n* = 6 per group).

Group	MLNs	Spleen	Liver	Blood	TR
Saline	0/6 (0.0%)	0/6 (0.0%)	0/6 (0.0%)	0/6 (0.0%)	0/24(0.0%)
FMT50	1/6 (16.7%)	0/6 (0.0%)	0/6 (0.0%)	0/6 (0.0%)	1/24(4.2%)
FMT150	2/6 (33.3%)	1/6 (16.7%)	1/6 (16.7%)	0/6 (0.0%)	4/24(16.7%) *
FOLFOX	2/6 (33.3%)	1/6 (16.7%)	2/6 (33.3%)	0/6 (0.0%)	5/24(20.8%) *
FOLFOX+FMT50	1/6 (16.7%)	2/6 (33.3%)	0/6 (0.0%)	0/6 (0.0%)	3/24(12.5%)
FOLFOX+FMT150	1/6 (16.7%)	2/6 (33.3%)	1/6 (16.7%)	0/6 (0.0%)	4/24(16.7%) *

The numbers in the table represent the positive number of cultures of various organs and the translocation ratios (TR) in all groups. * *p* < 0.05 vs. saline group at day 6. MLNs: mesenteric lymph nodes.

**Table 3 ijms-21-00386-t003:** *p*, F values and false discovery rate (FDR) correction from tests of Bray–Curtis and Jaccard similarity indices to evaluate the significance of FMT in changing the gut microbiota from the stool of subcutaneously injected colorectal cancer mice challenged with FOLFOX (*n* = 5 for each group).

		Bray-Curtis			Jaccard		
		F	*p*	FDR Corrected *p* Values	F	*p*	FDR Corrected *p* Values
Saline	FMT50	2.15145	0.00794	0.009528	1.88344	0.00794	0.011910
Saline	FOLFOX	2.94918	0.03968	0.039680	2.37435	0.03968	0.039680
FMT50	FOLFOX	4.16423	0.00794	0.009528	2.94634	0.00794	0.011910
Saline	FOLFOX+FMT50	8.49609	0.00794	0.009528	5.64205	0.00794	0.011910
FMT50	FOLFOX+FMT50	10.77795	0.00794	0.009528	6.44849	0.00794	0.011910
FOLFOX	FOLFOX+FMT50	2.73125	0.00794	0.009528	2.35479	0.01587	0.019044

FMT50: 50 mg/mL. FDR: false discovery rate. If *p* < 0.05, differences were considered significant.

**Table 4 ijms-21-00386-t004:** Primers used for real-time PCR.

Gene	Forward Primer (5′→3′)	Reverse Primer (5′→3′)
*ZO-1*	ACCCGAAACTGATGCTGTGGATAG	GTGGTTGTCACCAGCATCAG
*Occludin*	ATGTCCGGCCGATGCTCTC	TTTGGCTGCTCTTGGGTCTGTAT
*Claudin 2*	ATGCCTTCTTGAGCCTGCTT	AAGGCCTAGGATGTAGCCCA
*JAM-A*	CTGATCTTTGACCCCGTGAC	ACCAGACGCCAAAAATCAAG
*TLR1*	TCAAGCATTTGGACCTCTCCT	TTCTTTGCATATAGGCAGGGC
*TLR2*	ACAATAGAGGGAGACGCCTTT	AGTGTCTGGTAAGGATTTCCCAT
*TLR4*	ATGGCATGGCTTACACCACC	GAGGCCATTTTTGTCTCCACA
*TLR5*	AGC CTC CGC CTC CAT TCT TC	TCA CGC CTC TGA AGG GGT TC
*TLR6*	TGAGCCAAGACAGAAAACCCA	GGGACATGAGTAAGGTTCCTGTT
*MYD88*	ATCGCTGTTCTTGAACCCTCG	CTCACGGTCTAACAAGGCCAG
*GAPDH*	TGAACGGGAAGCTCACTGG	TCCACCACCCTGTTGCTGTA

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
