# Peer review of "Fecal Microbiota Transplantation Prevents Intestinal Injury, Upregulation of Toll-Like Receptors, and 5-Fluorouracil/Oxaliplatin-Induced Toxicity in Colorectal Cancer"

_ijms, 2020, doi:10.3390/ijms21020386_

Round 1
Reviewer 1 Report
“Fecal microbiota transplantation prevents intestinal injury and toll-like receptors upregulation from 5-fluorouracil/oxaliplatin toxicity in colorectal cancer” by Chang et al is a very interesting and original scientific manuscript.
The manuscript covers a contemporary topic of fecal microbiota transplantation, confirming its benefits in reducing side-effects of chemotherapy in colorectal cancer animal models. The text is well written, clear and easy to read. The impressively broad set of different experiments performed by the authors are explained in detail. The results are presented very well and all the key messages conveyed to the reader in discussion chapter in a thorough manner.
Minor revisions:
- 2.1. section title is a bit long and confusing
- Majority of figure and table captions include sentences that describe experimental procedures already covered in materials and methods. These repetitions are not necessary (i.e. lines 123-126, 175-177, 204-205, etc.)
- All the figures have xa, xb, xc in their artwork. Since the images of the same figure are grouped, the individual images should be named only a, b , c, etc (instead of 1a, 1b, 1c for Figure 1, etc.).
- The text often mentions control and FOLFOX groups. I presume control groups are 1, 2 and 3, while FOLFOX groups are 4, 5 and 6. This should somehow be noted in the table 1.
- Figure 1b – x-axis should be labeled as “Days after treatment”
- Figure 6 – (A), (B), (C), etc is missing from the figure caption.
- Figure 7 – graphs should note FMT50 and FOLFOX + FMT50 to indicate the 50mg/ml.
- Line 334 – this probably refers to figure 7D not 6D?
- Table 3 – are the P-values corrected for multiple testing? All the values presented in the table show significant changes. Please elaborate and add to 4.11 Statistical analysis section.
- Line 378 – “An improved level of toxicity” implies higher level of toxicity. “For decreasing level of toxicity” should be better.
- Discussion section is 4 pages long. The authors should consider making this section more concise.
- 4.10.2 Sequencing and analysis of 16S rDNA chapter should elaborate the processing of amplicon sequences. Also, how is it possible to determine a-diversity via one-way ANOVA? Perhaps ANOVA was used to compare a-diversity between groups?
- The authors should consider rewriting conclusions chapter to make it more clear.
Author Response
Dec 8, 2019
Editor-in-Chief
International Journal of Molecular Sciences
Dear Editor and Reviewers:
We thank you for the letter and the comments on our manuscript titled “Fecal microbiota transplantation prevents intestinal injury and toll-like receptors upregulation from 5-fluorouracil/oxaliplatin toxicity in colorectal cancer.” (ID: 651973). The paper was coauthored by Ching-Wei Chang, Hung-Chang Lee, Li-Hui Lee, Jen-Shiu Chiang Chiau, Tsang-En Wang, Wei-Hung Chuang, Ming-Jen Chen, Horng-Yuan Wang, Shou-Chuan Shih, Chia-Yuan Liu, and Tung-Hu Tsai. All the comments were insightful and have helped in considerably improving our manuscript. We have carefully addressed the issues raised by the reviewers, and the point-by-point responses to their comments are attached herewith.
Intestinal mucositis is a common adverse effect associated with the use of FOLFOX for treating colorectalcancer patients. Its occurrence results in an increase in the hospitalization duration and infection risk, and a reduction in the levels of anti-neoplastic agents used for treatment. This in turn is attributable to a reduced survival rate and substantial burden on Medicare. Thus, we sought to develop a convenient and novel method to alleviate mucositis, by investigating the effects of a fecal microbiota transplant (FMT) on FOLFOX-induced mucosal injury in BALB/cmice implanted with syngeneic CT26 colorectal adenocarcinoma cells.We believe that our study makes a significant contribution to the literature, becauseour results show that FMT safely reduces the severity of intestinal mucositis and diarrhea, without affecting the anti-tumor effects of FOLFOX or causingbacteremia in colorectalcancer-bearing mice.
Further, we believe that this paper will be of interest to the readership of your journal because our results demonstrate that FMT can mitigate FOLFOX-induced intestinal toxicity in a safe manner. Specifically, FMT is able to heighten the survival rate of chemotherapy-treated mice and could be used in a novel therapeutic strategy for managing chemotherapy-induced mucositis, to improve the prognosis of colorectal cancer patients receiving chemotherapy in the future.
This manuscript has not been published or presented elsewhere in part or in entirety and is not under consideration by another journal. The study design was approved by the appropriate ethics review board. We have read and understood your journal’s policies, and we believe that neither the manuscript nor the study violates any of these. There are no conflicts of interest to declare.
Thank you for your consideration. I look forward to hearing from you.
Sincerely,
Yu-Jen Chen M.D., Ph.D.
Department of Radiation Oncology, Mackay Memorial Hospital,
No. 92, Section 2, Chung San North Road, Taipei 104, Taiwan.
Fax: (886) 2 2809 6180
Phone: (886) 2 2809 4661 ext. 2301
E-mail: chenmdphd@gmail.com
Point-by-point responses to the reviewers’ comments (Chen et al.)
Title: Fecal microbiota transplantation prevents intestinal injury and toll-like receptors upregulation from 5-fluorouracil/oxaliplatin toxicity in colorectal cancer
We sincerely thank the reviewers for the constructive criticisms and valuable comments, which were of considerable help in improving the manuscript. Our responses to the reviewers’ comments are provided below.
Reviewers' comments:
Reviewer #1:
Q.- 2.1. section title is a bit long and confusing
Response:We have shorten the 2.1. section title and make it more clearly in the revised manuscript (Results section) and as follows:
2.1. Effects and safety of FMT on colorectal cancer-implanted mice challenged with FOLFOX….
Q.- Majority of figure and table captions include sentences that describe experimental procedures already covered in materials and methods. These repetitions are not necessary (i.e. lines 123-126, 175-177, 204-205, etc.)
Response:We have deleted the sentences that describe experimental procedures already covered in materials and methods in the majority of figures and tables captions in the revised manuscript and as follows:
Table 1, Table 2, Table 3
Figure 2, Figure 3, Figure 4, Figure 5, Figure 6, Figure 7
Q.- All the figures have xa, xb, xc in their artwork. Since the images of the same figure are grouped, the individual images should be named only a, b , c, etc (instead of 1a, 1b, 1c for Figure 1, etc.).
Response:Thank you for the reviewer’s comment. We have revised the figure in in the revised manuscript and as follows:
Figure 1, Figure 3, Figure 4, Figure 5, Figure 6, Figure 7
Q.- The text often mentions control and FOLFOX groups. I presume control groups are 1, 2 and 3, while FOLFOX groups are 4, 5 and 6. This should somehow be noted in the table 1.
Response:Thank you for the reviewer’s comment. The control group mentioned in the text means saline group as an untreated control. To make this point more clear, we add "(untreated control)" in Table 1in the revised manuscript and as follows:
Table 1.Colorectal cancer-bearing mice (n = 10 in each group) groups in which the effects of FMTon FOLFOX-induced intestinal mucositis were evaluated
|
Groups |
Regimen |
|
1 |
Saline(untreated control) |
|
2 |
FMT50 |
|
3 |
FMT150 |
|
4 |
FOLFOX |
|
5 |
FOLFOX + FMT50 |
|
6 |
FOLFOX + FMT150 |
FMT50: 50 mg/mL, FMT150: 150 mg/mL
Q.- Figure 1b – x-axis should be labeled as “Days after treatment”
Response:Thank you for the reviewer’s comment. We have labeled as “Days after treatment” in Figure 1b in the revised manuscript and as follows:
Figure 1b
Q.- Figure 6 – (A), (B), (C), etc is missing from the figure caption.
Response:We have added (A), (B), (C), (D) in the figure 6 caption in the revised manuscript and as follows:
Figure 6. Serum protein levels of (A) Il-1β, (B) Il-6, (C) TNF-α, and (D) Il-10 in subcutaneously injected colorectal carcinoma mice exhibiting FOLFOX-induced intestinal damage with/without a FMT. ………….
Q.- Figure 7 – graphs should note FMT50 and FOLFOX + FMT50 to indicate the 50mg/ml.
Response:We have noted FMT50 and FOLFOX + FMT50 to indicate the 50mg/ml in the figure 7 in the revised manuscript and as follows:
.
Q.- Line 334 – this probably refers to figure 7D not 6D?
Response:It refer to figure 7D not 6D.We have changed the “Figure 6 D” to “Figure 7D” in the revised manuscript and as follows:
Result section 2.7: …and no significant differences were observed in the microbiota of the saline, FMT, FOLFOX, and FOLFOX+FMT groups (Figure 7D). However,……
Q.- Table 3 – are the P-values corrected for multiple testing? All the values presented in the table show significant changes. Please elaborate and add to 4.11 Statistical analysis section.
Response:Thank you for the reviewer’s comment. In Table 3, the community structure (β-diversity) was analyzed by performing permutational multivariate analysis of variance (PERMANOVA) of ranked Bray-Curtis and Jaccard distances, using the CLC Genomics Workbench 12 software (CLC Bio, USA). A PERMANOVA analysis for each pair of groups and the results of the test (pseudo-f statistic and p-value). Fisher’s Least Significant Difference (LSD) post hoc tests which correct for multiple testing were also shown.We have clarified and revised the Table 3and 4.11 Statistical analysis sectionin the revised manuscript and as follows:
4.11 Statistical analysis: ………. Data were analyzed using IBM SPSS software (version 21.0; SPSS Institute). Taxonomy (i.e., phyla and OTUs) and α-diversity were analyzed by one-way ANOVA. The community structure (β-diversity) was analyzed by performing permutational multivariate analysis of variance (PERMANOVA) of ranked Bray-Curtis and Jaccard distances, using the CLC Genomics Workbench 12 software (CLC Bio, USA). A PERMANOVA analysis for each pair of groups and the results of the test (pseudo-f statistic and p-value). Fisher’s Least Significant Difference (LSD) post hoc tests which correct for multiple testing were also shown.The results were considered to be statistically significant if P < 0.05.
Q.- Line 378 – “An improved level of toxicity” implies higher level of toxicity. “For decreasing level of toxicity” should be better.
Response: We have changed the “An improved level of toxicity” to “For decreasing level of toxicity” in the revised manuscript (Section: Discussion) and as follows:
….. Multiple tools exist, including dietary modifications, probiotics, and synthetically engineered bacteria, through which the microbiota may be modulated for decreasing level of toxicity associated with chemotherapy…….
Q.- Discussion section is 4 pages long. The authors should consider making this section more concise.
Response:Thank you for the reviewer’s comment. We have make the section of discussion more concise in the in the revised manuscript (Section: Discussion).
Q.- 4.10.2 Sequencing and analysis of 16S rRNA chapter should elaborate the processing of amplicon sequences. Also, how is it possible to determine a-diversity via one-way ANOVA? Perhaps ANOVA was used to compare a-diversity between groups?
Response: Thank you for the reviewer’s comment. (1) Amplicon sequences analysis was based on the functions performed by a CLC bio "OTU clustering". We should elaborate the processing of amplicon sequences in Sequencing and analysis of 16S rRNA chapter. (2) Taxonomy (i.e., phyla and OTUs) and α-diversity were determined by “alpha diversity” function, and test the difference via one-way ANOVA. We have clarified and revised these in the revised manuscript (Section: Materials and Method. Statistical analysis) and as follows:
Elaborate the processing of amplicon sequences:
4.10.2 Sequencing and analysis of 16S rRNA……… CLC Genomics Workbench 12 software (CLC Bio, USA) was used along with the Greengenes 16S rRNA Taxonomy Database (gg_13_8) to conduct 16S rDNA analysis. “OTU clustering” function was used with de novo OTU clustering default parameter. “OTU clustering” function will cluster similar reads base on 97% similarity. The minimal criteria of OTU is 10 as default. The OTU clusters will be identified and used to following analysis and data visualization.
4.11 Statistical analysis…………Data were analyzed using IBM SPSS software (version 21.0; SPSS Institute). Taxonomy (i.e., phyla and OTUs) and α-diversity were analyzed by one-way ANOVA. The community structure (β-diversity) was analyzed by performing permutational multivariate analysis of variance (PERMANOVA) of ranked Bray-Curtis and Jaccard distances, using the CLC Genomics Workbench 12 software (CLC Bio, USA). ……. The results were considered to be statistically significant if P < 0.05.
Q.- The authors should consider rewriting conclusions chapter to make it more clear.
Response:Thank you for reviewer’s comment. We have modified the sentences in Conclusions in the revised manuscript and as follows:
Conclusions…Our murine model of colorectal cancer with severe FOLFOX-induced intestinal mucositis exhibited changes in gut microbiota; the development of mucositis might be encouraged by the activation of the gut microbiota present downstream of the TLR signaling pathway, following MyD88 and NF-κB expression. NF-kB expression results in the generation of apoptotic signals and pro-inflammatory cytokines, which sequentially contribute to intestinal mucosal integrity. Through the modulation of the gut microbiota present downstream of the TLR signaling pathway and by generating pro-inflammatory responses, FMT mitigated FOLFOX-induced intestinal toxicity in a safe manner. FMT could safely ameliorate inflammation, protect the epithelium by maintaining intestinal epithelial integrity, and reduce the severity of mucositis following FOLFOX treatment. The possible mechanisms may involve the gut microbiota-TLR-MyD88-NF-kB signaling pathway in mice with implanted colorectal carcinoma cells. Specifically, FMT is able to heighten the survival rate of chemotherapy-treated mice and could be used in a novel therapeutic strategy for managing chemotherapy-induced mucositis, to improve the prognosis of patients receiving chemotherapy in the future..
Reviewer 2 Report
The study uses fecal microbiota transplants (FMT) to prevent chemotherpy-induced toxicity. The approach is nice but more exploration of the data and experiments are needed. I have gathered my concerns and suggestions below. I also suggest corrections to english language below
The title needs to be modified because it is unclear as it is. I suggest: "Fecal microbiota transplantation prevents intestinal injury, upregulation of toll-like receptors and 5-fluorouracil/oxaliplatin-induced toxicity in colorectal cancer" Abstract: modify the sentence starting at line 41 for not to repeat during twice. At line 43, remove further. Line 45 remove 'cells observed via' and 'staining' Abstract and thoughout the text: The authors cannot state NFKB activated or activity, because they have not studied activity. They may only refer to NFKB positive cells or to the expression of NFKB Line 46: I would finish the sentence after treatment and start new sentence, which should be modified accordingly
INTRODUCTION
The authors should open up in the first paragraph what are gut microbes and what they do Line 71 abbreviate FMT, and be consiostent thorughout the manuscipt. Either abbreviated always or not Line 78, reference missing after remission Line 80: How the mechansims for FMT can be unclear? Isn't it obvious that the gut microbiota is the underlying mechanism? Line 89 reference missing I suggest the authors shortly describe, which known effect 5_FU has on the gut microbiota Line 105 and 106 reference missing I suggest not including results at the final of the introduction
RESULTS
Each subtitle should begin with The Add information about group sizes to the results, figure legends and of course methods. It is difficult to estimate the statistical power Line 120 remove 'changes in' Line 127 add the before subcutaneously Line 129 add the before tumor, add by FOLFOX after prevented Line 135 remove 'and the results were compared' Line 142 I suggest wrting ' The survival was monitored for 30 days with Kaplan-Meier survival analysis.' Line 170, the authors cannot state that no bacteria were translocated to the bloodstream. For sure at some point there have been bacteria in the blood traveling to spleen and liver. They should state that at necropsy no bacteria were detected in blood. Line 181 replace on by at Did the authors use any other method to analyze apoptosis? Some other methods may be more powerful than TUNEL. Please comment Remove the sentence at line 248, and start a new sentence by There at line 249. Line 251 replace reactive by positive and line 253 replace activity by expression. No activity was measured. If the authors want to say something about activity, why not studying the localization of NFKB in cytosol vs nucleus? Please comment From figure 3 onwards, why is the FMT150 group excluded from the results? I strongly suggest including the results, or at least justify, why this group is not included in the analyses and figures chapter 2.5 correct the gene names or otherwise do not use italics for occludin and claudin TLR5 is one of the most important TLR in shaping up the microbiota composition. Moreover, a recent study in Cancers showed the importance of flagellin (that is recognized by TLR5) in colorectal cancer. The authors need to analyze TLR5 mRNA and include the results Line 309 replace 'regulation of expression' by 'levels Line 312-3 remove by assays Line 313 remove protein Line 316 replace expression by levels. IL6 is likely derived from adipose tissue and exported to serum, not expressed there Gut microbiota results: the authors should explore much more the data. They need to present the gut microbiota composition also at family and genus level. Which % of the sequences or total bacteria Firmicutes and Bacteroidetes represented. Phylum level names are not written in italics, only genus. Because the authors did not use antibiotics or bowel cleansing before FMT, they need to show how well did the FMT really persist in the recipient mice. Moreover, they did not work in anaerobic conditions, causing an important loss of anerobic bacteria. This is a limitation. Therefore the authors need to analyze the composition of the FMT and the fecal sample at necropsy. If not many bacteria persisted, the effect may have been purely placebo. Line 334 remove however Line 335 replace 'analysis was performed' by was analyzed' Line 336 replace composition by beta-diversity Line 337 remove the sentence FMT and FOLFOX.... Line 339 replace induced by affected Line 340 Replace as by because. Starting the sentence with as is incorrect english The authors should interpret the results of heat map and clustering better In figure 7d the title of the axis is missing For the first time in table 3 I see that the n per group is 4. For microbiota studies it is small size. Can the authors comment on this?
DISCUSSION
Line 367-8. Split the sentence in two Line 371 replace has by may have. This is only preclinical study and does not provide evidence for clinical studies Line 376 reference missing Line 378 reference missing Line 378 reference missing Line 380 abbreviate FMT! Line 384-5 How did it mimick colon cancer patients? Line 386 add the before intestinal Line 422 reported inconsistently, correct Add more references to line 430. There are more studies that should be acknowledged Line 434-5 in addition to chemotherapy, also cancer itself can cause bacterial translocation. Please add Line 436-7, replace as by because and remove extracts Line 422 add reference after pneumonia Line 444 remove the microbiome-based therapeutic interventions Line 446 add for instance before Li et al, correct FMT abbreviated Line 451 correct sentence related to bloodstream according to my comment above Line 464 TLRs do not only recognize bacterial ligands but also viral and fungi. Please correct Line 475 it is not necessary to repeat the different cytokines in parentheses. Add reference Line 483 references missing for the TLR ligands. Correct that it is the heterodimers formed by combinations of TLR1, 2, & 6 that can achieve the recognition of so many ligands. For instance, TLR6 does not alone recognize peptidoglycans The paragraph dealing with cytokines should be combined or at least follow the paragraph of toll-like receptors. Both are linked Line 510 remove the sentence starting with Oral FMT. It is useless. Replace again as by because
METHODS
Describe the animal procedures better. group size. Were they single-housed? If not, then the limitation is coprophagy > transferring the gut microbiota in the cage > the sample size has no power The sections should be reorganized and redivided. Startying from the animals and cell cultures, followed by tumor implantation, FMT and chemotherapy. Body weight and other physiological measures should be an own chapter chapter 4.5 the title should be Translocation of gut microbiota. The authors did not study infections Line 617 replace evaluation by analyses Line 621 remove wax Line 629 replace we used somewhere by was used chapter 4.9 start by RNA extraction, then cDNA synthesis then qPCR. Now it is mixed. In the two last sentences the same thing is said in two different ways. Correct Line 671 do not use we collected The authors should rationalize why the collected fecal sample and not colon content at necropsy. This is also a limitation. Colon content would be more represantitive of what is really happening inside the gut How long time DNA was stored at -30? It is not an optimal temperature Line 674 16S rRNA! There is no ribosomal DNA Remove thesentence strating at line 692 What was the rarefaction levels of the samples? How mny sequences? More information is needed Did the authors make FDR corrcetions. Please specify. Chapter 4.11 should start with the program with which the analyses were made
CONSCLUSIONS
the authors draw too bold conclusions. Please modify the sentences. In addition, use past time in the verbs. Remove NFKB activation as commented above.
Author Response
Dec 8, 2019
Editor-in-Chief
International Journal of Molecular Sciences
Dear Editor and Reviewers:
We thank you for the letter and the comments on our manuscript titled “Fecal microbiota transplantation prevents intestinal injury and toll-like receptors upregulation from 5-fluorouracil/oxaliplatin toxicity in colorectal cancer.” (ID: 651973). The paper was coauthored by Ching-Wei Chang, Hung-Chang Lee, Li-Hui Lee, Jen-Shiu Chiang Chiau, Tsang-En Wang, Wei-Hung Chuang, Ming-Jen Chen, Horng-Yuan Wang, Shou-Chuan Shih, Chia-Yuan Liu, and Tung-Hu Tsai. All the comments were insightful and have helped in considerably improving our manuscript. We have carefully addressed the issues raised by the reviewers, and the point-by-point responses to their comments are attached herewith.
Intestinal mucositis is a common adverse effect associated with the use of FOLFOX for treating colorectalcancer patients. Its occurrence results in an increase in the hospitalization duration and infection risk, and a reduction in the levels of anti-neoplastic agents used for treatment. This in turn is attributable to a reduced survival rate and substantial burden on Medicare. Thus, we sought to develop a convenient and novel method to alleviate mucositis, by investigating the effects of a fecal microbiota transplant (FMT) on FOLFOX-induced mucosal injury in BALB/cmice implanted with syngeneic CT26 colorectal adenocarcinoma cells.We believe that our study makes a significant contribution to the literature, becauseour results show that FMT safely reduces the severity of intestinal mucositis and diarrhea, without affecting the anti-tumor effects of FOLFOX or causingbacteremia in colorectalcancer-bearing mice.
Further, we believe that this paper will be of interest to the readership of your journal because our results demonstrate that FMT can mitigate FOLFOX-induced intestinal toxicity in a safe manner. Specifically, FMT is able to heighten the survival rate of chemotherapy-treated mice and could be used in a novel therapeutic strategy for managing chemotherapy-induced mucositis, to improve the prognosis of colorectal cancer patients receiving chemotherapy in the future.
This manuscript has not been published or presented elsewhere in part or in entirety and is not under consideration by another journal. The study design was approved by the appropriate ethics review board. We have read and understood your journal’s policies, and we believe that neither the manuscript nor the study violates any of these. There are no conflicts of interest to declare.
Thank you for your consideration. I look forward to hearing from you.
Sincerely,
Yu-Jen Chen M.D., Ph.D.
Department of Radiation Oncology, Mackay Memorial Hospital,
No. 92, Section 2, Chung San North Road, Taipei 104, Taiwan.
Fax: (886) 2 2809 6180
Phone: (886) 2 2809 4661 ext. 2301
E-mail: chenmdphd@gmail.com
Point-by-point responses to the reviewers’ comments (Chen et al.)
Title: Fecal microbiota transplantation prevents intestinal injury and toll-like receptors upregulation from 5-fluorouracil/oxaliplatin toxicity in colorectal cancer
We sincerely thank the reviewers for the constructive criticisms and valuable comments, which were of considerable help in improving the manuscript. Our responses to the reviewers’ comments are provided below.
Reviewer #2:
The title needs to be modified because it is unclear as it is. I suggest: "Fecal microbiota transplantation prevents intestinal injury, upregulation of toll-like receptors and 5-fluorouracil/oxaliplatin-induced toxicity in colorectal cancer"
Response:We have changed the title to "Fecal microbiota transplantation prevents intestinal injury, upregulation of toll-like receptors and 5-fluorouracil/oxaliplatin-induced toxicity in colorectal cancer" as your suggestion in the revised manuscript and as follows:
Article
Fecal microbiota transplantation prevents intestinal injury, upregulation of toll-like receptors and 5-fluorouracil/oxaliplatin-induced toxicity in colorectal cancer
ABSTRACT:
Abstract: modify the sentence starting at line 41 for not to repeat during twice.
Response: We have modified the sentence starting at Line 41 for not to repeat during twice in in the revised manuscript and as follows:
Abstract:……BALB/c mice implanted with syngeneic CT26 colorectal adenocarcinoma cells were orally administered FMT daily during and 2 days after 5-day injection of FOLFOX regimen, for 7 days…….
At line 43, remove further.
Response: We have removed “further” at Line 43 in the revised manuscript and as follows:
Abstract:…..Administration of FOLFOX significantly induced marked levels of diarrhea and intestinal injury. Further, FMT reduced the severity of diarrhea and intestinal mucositis. Additionally, the number of goblet cells…..
Line 45 remove 'cells observed via' and 'staining'
Response:We have removed 'cells observed via' and 'staining' at Line 45 in the revised manuscript and as follows:
Abstract:…..Additionally, the number of goblet cells and zonula occludens-1 decreased, while apoptotic…..
Abstract and thoughout the text: The authors cannot state NFKB activated or activity, because they have not studied activity. They may only refer to NFKB positive cells or to the expression of NFKB Line 46: I would finish the sentence after treatment and start new sentence, which should be modified accordingly
Response:We have changed the state “NFKB activated or activity” to “NFKB positive cells or to the expression of NFKB” in the revised manuscript (abstract and throughout the text), and also have finished the sentence after treatment and started new sentence and as follows:
Throughout the text (Section: Introduction, Results, Discussions, and Conclusions):……NF-κB-activatedpositivecells…., …NF-κB activationexpression….., NF-κB activityexpression….., Activated NF-κB expression
Abstract……Additionally, the number of goblet cells and zonula occludens-1 decreased, while apoptotic and NF-κB-positivecells increased following FOLFOX treatment. The expression of toll-like receptors, MyD88 and serum IL-6 were upregulated following FOLFOX treatment………
INTRODUCTION:
The authors should open up in the first paragraph what are gut microbes and what they do
Response:Thank you for reviewer’s comment. We have open up in the first paragraph about microbes and what they do in the revised manuscript and as follows:
Introduction
Growing evidence not only implies that chemotherapeutics affect the intestinal microbial composition, but also that multidirectional interactions between the gut microbiota and host immune system may influence the development and progression of chemotherapy-induced intestinal inflammation [4-6].
Gastrointestinal toxicity is a severe, dose-limiting, toxic side effect of chemotherapeutics in cancer patients. Mucositis and diarrhea are a primary outcome of intestinal toxicity………………
Line 71 abbreviate FMT, and be consiostent thorughout the manuscipt. Either abbreviated always or not
Response:We have abbreviated FMT at Line 71 and consistent throughout the manuscript in the revised manuscript and as follows:
Introduction……Fecal microbiota transplant (FMT) is the transfer of fecal material, including bacteria and natural anti-bacterials, from a healthy individual into a diseased recipient. The manipulation of gut microbiota by fecal microbiota transplantationFMThas been performed in various disease models and clinical trials, and has received much public attention.
Throughout the manuscript…… fecal microbiota transplantationFMT……..
Line 78, reference missing after remission
Response:Thank you for the reviewer’s comment. We have added references (Smits, L.P et al.,. Gastroenterology 2013, 145, 946-953. Ooijevaar, R.E. et al., Annu. Rev. Med. 2019, 70, 335-351.) after remission at Line 78in the revised manuscript and as follows:
Introduction……individuals to patients with inflammatory bowel disease or irritable bowel syndrome resulted in clinical improvement and remission [9,10]. It has been also suggested that FMT shows promising therapeutic potential…….
Line 80: How the mechansims for FMT can be unclear? Isn't it obvious that the gut microbiota is the underlying mechanism?
Response:Thank you for the reviewer’s comment. Gut microbiota is the underlying mechanism for FMT. We have clarified revised the sentence at Line 80 in the revised manuscript and as follows:
Introduction……Although the underlying molecular and cellularmechanism for FMT remains unclear, it may involve the direct interaction of donor gut microbiota with that of the host, which subsequently mediates the effects observed on the host physiology, gut mucosal barrier and immune system……..
Line 89 reference missing
Response:Thank you for the reviewer’s comment. We have added references (Cammarota, G.et al., Gut 2017, 66, 569-580. Xu, M.Q. et al., World J. Gastroenterol. 2015, 21, 102-111.) at Line 89in the revised manuscript and as follows:
Introduction……Further, transient adverse events, including mild fever, abdominal pain, diarrhea, exhaustion, flatulence, and fatigue were reported following FMT [13,14]. These adverse effects were self-limiting;…….
I suggest the authors shortly describe, which known effect 5_FU has on the gut microbiota
Response:Thank you for the reviewer's comment. We have shortened the description about the effect 5-FU on the gut microbiota in the revised manuscript and as follows:
Introduction……Regimens based on 5-fluorouracil (5-FU),an anti-metabolite, anti-cancer agent, have been the first-choice chemotherapy regimens used for colorectal cancer patients [18]. Among regimens based on 5-FUwith the cytotoxic agent oxaliplatin, FOLFOX (5-fluorouracil, leucovorin, and oxaliplatin) has been widely used in standard chemotherapy for advanced and metastatic colorectal cancer treatment [18,19]……In our previous study, we reported on a colorectal cancer murine model with FOLFOX-induced intestinal mucositis that may enable us to effectively investigate the mechanism underlying intestinal injury, and its possible interaction with potential therapeutics [21]. The development of FOLFOX-induced mucositis involves changes in gut microbiota….By modulating the gut microbiota and pro-inflammatory responses, probiotics Lactobacillus casei variety rhamnosus mitigated FOLFOX-induced mucositis [21].;…….
Line 105 and 106 reference missing
Response:Thank you for the reviewer’s comment. We have added references (Chang, C.W et al., Front. Microbiol. 2018, 9, 983.) at Line 105 and 106in the revised manuscript and as follows:
Introduction……which contribute to gastrointestinal injury [21]. By modulating the gut microbiota and pro-inflammatory responses, probiotic Lactobacillus casei variety rhamnosus mitigated FOLFOX-induced mucositis [21].…….
I suggest not including results at the final of the introduction
Response:We have deleted the results at the final of the introduction in the revised manuscript and as follows:
Introduction…….Accordingly, in the current study we sought to investigate the effect and safety of FMT on 5-FU-based chemotherapy (FOLFOX)-induced intestinal mucosal injury in mice with colon cancer. We performed a FMT using stool from healthy wild-type donor mice to restore microbial diversity and composition. Our results revealed that FMT did not affect the anti-tumor effects of FOLFOX. FMT could safely ameliorate inflammation, protect the epithelium by maintaining intestinal epithelial integrity, and reduce the severity of mucositis following FOLFOX treatment. The possible mechanisms of action for FMT were also elucidated.
RESULTS
Each subtitle should begin with The Add information about group sizes to the results, figure legends and of course methods. It is difficult to estimate the statistical power
Response:We have make each subtitle begin with ”The” and also add “group sizes” to the results, figure legends and methods in the revised manuscript and as follows:
Results…2.1 Theeffects of……, 2.2The effects of….., 2.3 Theeffects of…., 2.4 The effects of ….., 2.5 Theeffects of ……, 2.6 Theeffects of ….., 2.7 Theeffects of ……
Results…Figure 3…(n = 6 for each group), Figure 4…(n = 6 for each group), Figure 5…(n = 6 for each group), Figure 6…(n = 6 for each group), Figure7…(n = 5 for each group), Table 3… (n = 5 for each group),
Line 120 remove 'changes in'
Response:We have removed “change in “ in the revised manuscript and as follows:
Results……. FMT did not affect changes inbody weight in the saline and FOLFOX groups (Figure 1B)……..
Line 127 add the before subcutaneously
Response:We have add “the’ before subcutaneously in the revised manuscript and as follows:
Results…. The size ofthesubcutaneously injected tumors was recorded…….
Line 129 add the before tumor, add by FOLFOX after prevented
Response:We have add “the’ before tumor, add “by FOLFOX” after prevented in the revised manuscript and as follows:
Results….thetumor growth was significantly preventedby FOLFOX,…….
Line 135 remove 'and the results were compared'
Response:We have removed 'and the results were compared' in the revised manuscript and as follows:
Results….Diarrhea scores for all groups were recorded daily and the results were compared,…….
Line 142 I suggest wrting ' The survival was monitored for 30 days with Kaplan-Meier survival analysis.'
Response:We have changed “Survival was monitored for 30 days. A Kaplan–Meier survival analysis of the mice was performed” to “The survival was monitored for 30 days with Kaplan-Meier survival analysis” as your suggestion in the revised manuscript and as follows:
Results….Survival was monitored for 30 days. A Kaplan–Meier survival analysis of the mice was performed. The survival was monitored for 30 days with Kaplan-Meier survival analysis,…….
Line 170, the authors cannot state that no bacteria were translocated to the bloodstream. For sure at some point there have been bacteria in the blood traveling to spleen and liver. They should state that at necropsy no bacteria were detected in blood.
Response:Thank you for the reviewer’s comment. We cannot state that no bacteria were translocated to the bloodstream in our manuscript at Line 170. We have clarified and revised the sentence at Line 170 as your suggestion in the revised manuscript and as follows:
Results….urthermore, the total translocation ratio was significantly higher in the FMT150, FOLFOX, and FOLFOX+FM150 groups, however, not in the FMT50 and FOLFOX+FMT50 groups, and was higher than that for the saline group (p< 0.05). At necropsy, no bacteria were detected to the bloodstream in any of the groups(Table 2)…….
Line 181 replace on by at
Response:We have replaced “on” by “at” in the revised manuscript and as follows:
Results….As expected, FOLFOX induced substantial histological changes in the intestinal mucosal layer onatday 6 (Figure 2A),…….
Did the authors use any other method to analyze apoptosis? Some other methods may be more powerful than TUNEL. Please comment
Response:Thank you for the reviewer’s comment. In our previous investigations (Chang, C.W et al., Front. Microbiol. 2018, 9, 983), we noted that FOLFOX-induced mucositis in mice with colon cancer is mainly mediated by the apoptotic pathway. FOLFOX administration caused a marked increase in TUNEL-positive apoptotic cells in the intestinal crypts. To elucidate the role of apoptotic cascades, the expression of caspase-8, BAX, and BCL-2 proteins was measured. That conclusion was drawn by assessing TUNEL, Bax, Bcl-2 and caspase 8 with consistent profiles of changes.
Therefore, in the current study, we sought to investigate the effect and safety of FMT on this FOFLOX-induced intestinal mucosal injury in mice model. TUNEL assay was performed to verifying the apoptosis activity of this mice model. The related mechanism will be explored further in the future.
Reference:Chang, C.W.; Liu, C.Y.; Lee, H.C.; Huang, Y.H.; Li, L.H.; Chiau, J.C.; Wang, T.E.; Chu, C.H.; Shih, S.C.; Tsai, T.H., et al. Lactobacillus casei Variety rhamnosus Probiotic Preventively Attenuates 5-Fluorouracil/Oxaliplatin-Induced Intestinal Injury in a Syngeneic Colorectal Cancer Model. Front. Microbiol. 2018, 9, 983.
Remove the sentence at line 248, and start a new sentence by There at line 249.
Response:We have removed the sentence at line 248 and and start a new sentence by There at line 249 in the revised manuscript and as follows:
Results….An antibody against the NF-κB p65 subunit was used for immunohistochemical analysis and it revealed that there.Therewere many p65-reactive cells in the crypts of FOLFOX-treated intestine in jejunal segments,,……
Line 251 replace reactive by positive and line 253 replace activity by expression. No activity was measured. If the authors want to say something about activity, why not studying the localization of NFKB in cytosol vs nucleus?
Response:Thank you for reviewer’s comment. No activity of NF-κB was measured in our study. We have replaced “reactive” by “positive” at Line 251 and “activity” by “expression” at Line 253in the revised manuscript and as follows:
Results….There were many p65-reactive cells in the crypts of FOLFOX-treated intestine in jejunal segments (32.94 ± 2.2vs. 2.24 ± 0.21 cells/crypt, p < 0.001, Figure 3Aand 3G), as compared to the saline group. FMT50 significantly reduced the FOLFOX-induced increase in the number of p65-positivecells in the intestine (10.55 ± 1.27vs. 32.94 ± 2.2 cells/crypt, p < 0.001, Figure 3Aand 3G). These findings indicated that FOLFOX induced NF-κB expression, while FMT inhibited FOLFOX-induced NF-κB expressionacross the crypts of jejunal segments.,,……
Please comment From figure 3 onwards, why is the FMT150 group excluded from the results? I strongly suggest including the results, or at least justify, why this group is not included in the analyses and figures
Response:Thank you for the reviewer’s comment.
In our study, on day 6,the severity of diarrheawas clearly attenuated in mice treated with FMT50 and FMT150 in the FOLFOX groups (2.0 ± 0.15 and 2.1 ± 0.1 vs. 2.8 ± 0.13, p < 0.005; Figure 1D).The survivalwas monitored for 30 days with Kaplan-Meier survival analysis. FMT50 and FMT150 administration statistically improved survival in the FOLFOX group, as shown by the results of the log-rank test (p < 0.005, Figure 1E). With regard to thesafety of FMT(Table 2). Mesenteric lymph node, liver, and spleen samples showed evidence of translocation more frequently in the FMT150 group than in the FMT50 group, as compared to those of the saline group. Furthermore, the total translocation ratio was significantly higher in the FMT150 and FOLFOX+FM150 groups, however, not in the FMT50 and FOLFOX+FMT50 groups, and was higher than that for the saline group (p < 0.05). FMT150 excluded from the results is for safety concerned. Thus, an oral FMT consisting of an optimal dose of 50 mg/mL, proved safe and effective for reducing FOLFOX-induced intestinal mucosal injury in colorectal carcinoma-implanted mice. We have revised thesein the revised manuscript and as follows:
Results….Thus, an oral FMT consisting of a 50 mg/mL dose, proved safe and effective for reducing FOLFOX-induced intestinal mucosal injury in colorectal carcinoma-implanted mice; this was determined after measuring the villus height (344.16 ± 4.24.,,……
chapter 2.5 correct the gene names or otherwise do not use italics for occludin and claudin
Response:We have corrected the gene names not using italics for occludin and claudin at chapter 2.5 in the revised manuscript and as follows:
Results…. mRNA expression of ZO-1, Occludin, Claudin-2, and JAM-A in the jejunum of mice treated with FOLFOX were determined. FMT50 treatment alone did not markedly affect the mRNA expression of ZO-1, Occludin, Claudin-2, and JAM-A (Figure 4 A-D). ZO-1 mRNA expression ……Claudin-2 mRNA expression was markedly upregulated in the jejunal tissues…... FMT50 treatment significantly downregulated FOLFOX-induced Claudin-2 mRNA expression in jejunal tissues (0.22 ± 0.05 vs. 3.02 ± 1.43, p = 0.001; Figure 4 C).……
TLR5 is one of the most important TLR in shaping up the microbiota composition. Moreover, a recent study in Cancers showed the importance of flagellin (that is recognized by TLR5)in colorectal cancer. The authors need to analyze TLR5 mRNA and include the results
Response:Thank you for reviewer’s comment. TLR5 is one of the most important TLR in shaping up the microbiota composition. Moreover, a recent study in Cancers showed the importance of flagellin (that is recognized by TLR5) in colorectal cancer (Pekkala, S.et al., Blocking Activin Receptor Ligands Is Not Sufficient to Rescue Cancer-Associated Gut Microbiota-A Role for Gut Microbial Flagellin in Colorectal Cancer and Cachexia? Cancers (Basel) 2019, 11.).
We have analyzed the TLR5 mRNA expression in our study. It revealed that FOLFOX treatment significantly increased the mRNA expression of TLR5 (81.49 ± 21.47 vs. 1.00 ± 0.21, p < 0.001)in the jejunum of mice. The administration of FMT50 significantly reduced the expression levels of TLR5 (0.79 ± 0.28 vs. 81.49 ± 21.47, p < 0.001), ascompared to the levels in the FOLFOX group (figure as follow). We have included it in the revised manuscript (Result, Figure, Discussion and Materials and Methods section), also inserted reference (Pekkala, S.et al., Cancers (Basel) 2019, 11.) in the revised manuscript and as follows:
Figure 5E
Results…….Previous studies have shown that intestinal epithelial cells express several TLRs, including TLR1, TLR2, TLR4,TLR5,and TLR6, in vitroand in vivo [22.23]. Figures 5A-5Eshow the different expression patterns of TLR1, TLR2, TLR4,TLR5,and TLR6mRNA in the jejunum of mice. FOLFOX treatment significantly increased the mRNA expression of TLR1 (101.00 ± 15.60 vs. 1.00 ± 0.40, p < 0.001), TLR2(96.85 ± 17.10 vs. 1.00 ± 0.26, p < 0.001), TLR4(110.96 ± 23.83 vs. 1.00 ± 0.32, p < 0.001), TLR5 (81.49 ± 21.47 vs. 1.00 ± 0.21, p < 0.001),and TLR6 (171.47 ± 27.09 vs. 1.00 ± 0.35, p < 0.001)in the jejunum of mice. The administration of FMT50 significantly reduced the expression levels of TLR1 (0.26 ± 0.08 vs. 101.00 ± 15.60, p < 0.001), TLR2(0.59 ± 0.33 vs. 96.85 ± 17.10, p < 0.001), TLR4(0.30 ± 0.04 vs. 110.96 ± 23.83, p < 0.001), TLR5 (0.79 ± 0.28 vs. 81.49 ± 21.47, p < 0.001),and TLR6(0.13 ± 0.35 vs. 171.47 ± 27.09, p < 0.001), ascompared to the levels in the FOLFOX group. However, no significant difference was detected between the saline and FMT groups. Furthermore, FOLFOX treatment significantly increased the mRNA expression of MyD88 (3.88 ± 0.55 vs. 1.00 ± 0.24, p < 0.005, Figure 5 F)in the jejunum of mice. The administration of FMT50 significantly reduced the expression levels of MyD88 (0.52 ±0.1vs. 3.88 ± 0.55, p < 0.001, Figure 5 F), ascompared to those of the FOLFOX group.
Figure 5. mRNA expression levels of toll-like receptors and MyD88 in the jejunum of subcutaneously injected colorectal cancer mice exhibiting FOLFOX-induced intestinal damage with/without a FMT (n = 6 for each group). Relative mRNA expression levels of (A) TLR1, (B) TLR2, (C) TLR3, (D) TLR4, (E) TLR5,and (F) MyD88 were determined by Qpcr in jejunum tissues. FMT50: 50 mg/ml. Values are presented as the mean ± SEM. *: p< 0.05, **: p < 0.01, ***: p < 0.001.
Discussions……Previous studies have shown that intestinal epithelial cells expressed several TLRs, including TLR1, TLR2, TLR4, TLR5,and TLR6 in vitro and in vivo, and that these TLRs were expressed on the cell surface and mainly recognized microbial membrane components, such as lipids, lipoproteins, and proteins………TLR4 recognizes bacterial lipopolysaccharide (LPS). TLR5recognizes bacterial flagellin [48]. The heterodimers formed by combinations of TLR1, TLR2, and 6 recognizes a wide variety of pathogen-associated molecular patterns, including lipoproteins, peptidoglycans, lipoteichoic acids, zymosan, mannan, and tGPI-mucin [22,23,44,47].. In our study, FOLFOX treatment…….……..…In our study, FOLFOX treatment might disturb the original balance of gut microbiota and significantly upregulate the mRNA expression of TLRs (TLR1, TLR2, TLR4, TLR5, and TLR6) and key signal molecule MyD88………….
Line 309 replace 'regulation of expression' by 'levels
Response:We have replaced “regulation of expression” by “levels” in the revised manuscript and as follows:
Results….2.5 The effects of FMT on the regulation of mRNAexpressionlevels of tight junction proteins and toll-like receptors in the jejunum of colorectal cancer-bearing mice challenged with FOLFOX…….
Line 312-3 remove by assays
Response:We have removed “by assays” at Line 312-3 in the revised manuscript and as follows:
Results….After euthanasia, the effects of FMT treatment on serum levels of cytokines interleukin-1β (IL-1β), IL-6, tumor necrosis factor-α (TNF-α), and IL-10 in FOLFOX-treated mice were determined by assays,…….
Line 313 remove protein
Response:We have removed “protein” at Line 313 in the revised manuscript and as follows:
Results….the results of which are shown in Figures 6A-6D. IL-1β and IL-6 proteinlevels were markedly increased,…….
Line 316 replace expression by levels. IL6 is likely derived from adipose tissue and exported to serum, not expressed there
Response:We have replaced “expression” by “levels” at Line 316 in the revised manuscript and as follows:
Results….FMT treatment significantly suppressed IL-6 levels in the FOLFOX group (14.30 ± 2.26 vs. 24.73 ± 5.15, p < 0.05; Figure 6B)…….
Gut microbiota results: the authors should explore much more the data. They need to present the gut microbiota composition also at family and genus level.
Response:Thank you for the reviewer’s comment. Taxonomic analysis at the family and genus level also revealed that FOLFOX changed the composition of the gut microbiota. FMT altered this composition in the FOLFOX-challenged group, as compared to that in the saline group. We have explored the gut microbiota composition at the family and genus level in the figure as follows and in the revised manuscript and as follows:
Results….Taxonomic analysis at the phylum level revealed that FOLFOX changed the composition of the gut microbiota. FMT altered this composition in the FOLFOX-challenged group, as compared to that in the saline group (Figure 7A and 7B). Taxonomic analysis at the family and genus levels also showed the same trends of the FMT effects (data not shown)..….
Figure Changes infecal gut microbiota of subcutaneously injected colorectal cancer mice presenting FOLFOX-induced intestinal damagewith/without a FMT (n = 5 for each group). The composition of gut microbiota was determined by Bar charts of the relative abundance of gut microbiota at the family and genus level,
Which % of the sequences or total bacteria Firmicutes and Bacteroidetes represented.
Response:Thank you for the reviewer’s comment. Firmicutes and Bacteroidetes were the two major phyla and represented relative abundance (%) in all groups. We have clarified and revised this in the revised manuscript (Result section and Figure) and as follows:
Results: ….. Bacteroidetes(B)and Firmicutes (F) were identified as the major phyla present in all groups. FOLFOX markedly increased the relative abundance of F/B ratio, as compared to that of the saline control. These effects were abrogated by FMT50 administration in FOLFOX-injected mice (0.3 ± 0.04 vs. 0.73 ± 0.05 F/B ratio, p < 0.01; Figure 7C).
Figure 7. Changes in fecal gut microbiota ….(C) Abundance ofFirmicutes-to-Bacteroidetes (F/B) ratio….
Phylum level names are not written in italics, only genus.
Response:We have phylum level names not written in italics in the revised manuscript and as follows:
Results….The composition of gut microbiota in stool was evaluated by next-generation sequencing. Taxonomic analysis at the phylum level revealed that …….. (Figure 7A and 7B). Bacteroidetes and Firmicuteswere identified as the major phyla present in all groups. FOLFOX markedly increased the F/B ratio, as compared to that of the saline control. These effects were abrogated by FMT50 administration in FOLFOX-injected mice (0.3 ± 0.04 vs. 0.73 ± 0.05 F/B ratio, p < 0.01; Figure 7C).…….
Because the authors did not use antibiotics or bowel cleansing before FMT, they need to show how well did the FMT really persist in the recipient mice.
Response:
Note: Thank you for reviewer’s comment. We did not use antibiotics or bowel cleansing before FMT in this mice model. Aim of our study is to evaluate the“effect” of oral FMT on this FOLFOX-induced intestinal mucositis in colorectal cancer-bearing syngeneic mice model. We would like to show how well did the oral FMT really persist the effects on or in the recipient mic as follow:
1.FMT significantly amelioratedFOLFOX-induced severe diarrhea, bacterial translocation,and intestinal mucosal injury,and improved long-term survival associated with FOLFOX administration in CT26 colorectal cancer-bearing mice. 2.FMT significantly reducedFOLFOX-induced intestinalmucosal inflammation , apoptosis, and barrier integrity disruption.
3.FMT significantly amelioratedFOLFOX-inducedTLR expression and changes in fecal gut microbiota by analysis of Taxonomy (i.e., phyla and OTUs), α-diversity, and β-diversity
Accordingly, these results indicate that FMT has safe effect on or in the recipient mice model, and may have more relevant to future clinical applicationfor the management of chemotherapy-induced intestinal dysbiosis and toxicity.
Moreover, they did not work in anaerobic conditions, causing an important loss of anerobic bacteria. This is a limitation. Therefore the authors need to analyze the composition of the FMT and the fecal sample at necropsy.If not many bacteria persisted, the effect may have been purely placebo.
Response:Thank you for the reviewer’s comment.
(1) : For preparation of “fresh”fecal material, anaerobic storage and processing should be applied if possible (Cammarota, G.; et al.. Gut 2017, 66, 569-580). We did not work in anaerobic conditions for preparation of “frozen” fecal material. The fecal material was prepared according to previous study by Li, M. et al (Front. Microbiol. 2015, 6, 692.) . From aspect of gastroenterologists, FMT procedure is more clinical relevant (Cammarota, G.; et al.. Gut 2017, 66, 569-580). We have revised this point in the revised manuscript (Section: Discussion) and as follows:
Discussion…. Our study also has some limitations. We did not work in anaerobic conditions for preparation of frozen fecal material. The sample size of mice was small and mice were co-housed, not single-housed for this study. We did not conduct an analysis of the time-course and dose-dependent effects of FMT; it would have provided more information regarding their role in the pathogenesis of chemotherapy-induced mucositis..…….
(2)We have analyzed the composition of the FMT and the fecal sample (oral FMT50, fecal sample for FMT) at necropsy. Taxonomic analysis at the phylum level revealed that FOLFOX changed the composition of the gut microbiota (Figure, below). Oral FMT50 altered this composition in the FOLFOX-challenged group, as compared to that in the saline group (Figure, below). The gut microbiota composition of the fecal sample (oral FMT50) at the phylum level as shown in the Figure of Bar charts and Heatmap below. The composition of the fecal sample (oral FMT50, fecal sample for FMT) look different from FMT50, FOLFOX, FOLFOX+FMT50 group at necropsyand worth further investigation
Figure.Changes infecal gut microbiota of subcutaneously injected colorectal cancer mice presenting FOLFOX-induced intestinal damagewith/without a FMT. The composition of gut microbiota was determined by the following: Bar charts, Heatmap of the relative abundance of gut microbiota at the phylum level, Oral FMT50 (fecal sample for Transplantation of fecal microbiota)
Line 334 remove however
Response:We have removed “however” at Line 334 in the revised manuscript and as follows:
Results….and FOLFOX+FMT groups (Figure 7D). However, Beta diversity analysis was performed using the Jaccard and Bray-Curtis distance matrix analysis method,,…….
Line 335 replace 'analysis was performed' by was analyzed'
Response:We have replaced “analysis was performed” by “was analyzed” at Line 335 in the revised manuscript and as follows:
Results….Beta diversity analysis was performedwas analyzedusing the Jaccard and Bray-Curtis distance matrix analysis method, as illustrated in Table 3,….
Line 336 replace composition by beta-diversity
Response:We have replaced “composition” by “beta-diversity” at Line 336 in the revised manuscript and as follows:
Results….in Table 3, and it showed that there were significant differences in the compositionbeta diversity of the microbiota of the saline, FMT, FOLFOX, and FOLFOX+FMT groups (p < 0.05);,….
Line 337 remove the sentence FMT and FOLFOX....
Response:We have removed “he sentence FMT and FOLFOX....” at Line 337 in the revised manuscript and as follows:
Results….of the saline, FMT, FOLFOX, and FOLFOX+FMT groups (p < 0.05); FMT and FOLFOX treatment alone affected the composition of fecal gut microbiota. Furthermore, the oral administration of FMT specifically altered FOLFOX-affected gut microbiota in colorectal carcinoma-implanted mice (p < 0.01).…….
Line 339 replace induced by affected
Response:We have replaced “induced” by “affected” at Line 339 in the revised manuscript and as follows:
Results….the oral administration of FMT specifically altered FOLFOX-inducedaffected gut microbiota in colorectal carcinoma-implanted mice (p < 0.01).;,….
Line 340 Replace as by because. Starting the sentence with as is incorrect english
Response:We have replaced “as” by “because” at Line 340 in the revised manuscript and as follows:
Results….AsBecausea Jaccard analysis only accounts for the presence/absence of Operational taxonomic units (OTUs) and a Bray-Curtis analysis is more sensitive to changes in OTU abundance,,….
The authors should interpret the results of heat map and clustering better
Response:Thank you for the reviewer’s comment. We have interpret the results of heat map and clusteringin the revised manuscript and as follows:
Results….Taxonomic analysis at the phylum level revealed that FOLFOX changed the composition of the gut microbiota. FMT altered this composition in the FOLFOX-challenged group, as compared to that in the saline group (Figure 7A and 7B). Bacteroidetes (B) and Firmicutes (F) were identified as the major phyla present in all groups. FOLFOX changed the abundance of the Bacteroidetes and Firmicutes, as compared to that in the saline group (Figure 7B). FMT altered this abundance in the FOLFOX-challenged group (Figure 7B). FOLFOX markedly increased relative abundance the F/B ratio, as compared to that of the saline control.,….
In figure 7d the title of the axis is missing
Response: Thank you for the reviewer’s comment. We have add the title of the axis in figure 7d in the revised manuscript and as follows:
Results_Figure 7 D
.
For the first time in table 3 I see that the n per group is 4. For microbiota studies it is small size. Can the authors comment on this?
Response: Thank you reviewer’s comment. We corrected the number per group 4 to 5 in Table 3.This is the first study, as per our knowledge, to report on the potential and safety of the use of FMT for suppressing FOLFOX-induced mucositis in a colorectal cancer mouse model in vivo. FMT ameliorated FOLFOX-induced severe diarrhea, bacterial translocation, and intestinal mucosal injury, and improved long-term survival associated with FOLFOX administration in CT26 colorectal cancer-bearing mice. In this chemotherapy-induced intestinal toxicity animal model, microbiota studies with small size (n=5 per group) revealed FOLFOX-induced changes in fecal gut microbiota and ameliorated by FMT.Base on the 3R principle (Replace, Reduce, Refine) for animal experimentation, we kept the number of animal experiments as low as possible and only use the necessary number of animals for study. Lastly, the distress inflicted upon the animals is as low as possible.We have added this limitation in the revised manuscript (Table, Discussion) and as follows:
Table 3. P and F values from tests of Bray-Curtis and Jaccard similarity indices, for evaluating the significance of FMT for bringing about changes in the gut microbiota from the stool of subcutaneously injected colorectal cancer mice challenged with FOLFOX (n=5 for each group)
Discussion…. Our study also has some limitations. ………The sample size of mice was small and mice were co-housed, not single-housed for this study. We did not conduct an analysis of the time-course and dose-dependent effects of FMT………..
DISCUSSION
Line 367-8. Split the sentence in two
Response:We have split the sentence in two at Line 367-8 in the revised manuscript and as follows:
Discussion…. Furthermore, FMT reduced FOLFOX-induced intestinal mucosal inflammation and barrier integrity disruption; t. Thiswas characterized by immunohistological changes and expression of tight junction proteins.,….
Line 371 replace has by may have. This is only preclinical study and does not provide evidence for clinical studies
Response:We have replaced “has” by “may have” at Line 371 in the revised manuscript and as follows:
Discussion….Therefore, these results indicate that FMT hasmay haveclinical potential for the management of chemotherapy-induced intestinal dysbiosis and toxicity.,….
Line 376 reference missing
Response:Thank you for the reviewer’s comment. We have added references at Line 376in the revised manuscript and as follows:
Discussion……however, few studies have investigated the effect of FMT on gastrointestinal mucositis resulting from combined 5-FU and oxaliplatin chemotherapy in animal models [8,14,24-27].Multiple tools exist, including.…….
Line 378 reference missing
Response: Thank you for the reviewer’s comment. We have added reference (Alexander, J.L et al., Nat. Rev. Gastroenterol. Hepatol. 2017, 14, 356-365.) at Line 378in the revised manuscript and as follows:
Discussion……Multiple tools exist, including dietary modifications, probiotics, and synthetically engineered bacteria, through which the microbiota may be modulated for decreasing level of toxicity associated with chemotherapy[24]. However, the evidence supporting the use of FMT..…….
Line 380 abbreviate FMT!
Response:We have abbreviated FMT at Line 380 in the revised manuscript and as follows:
Discussion…. Li et al revealed that fecal transplantationFMT from healthy mice might alleviate weight loss, colon shortening, and intestinal mucositis induced by 5-FU in healthy BALB/c mice [28].….
Line 384-5 How did it mimick colon cancer patients?
Response: Thank you for the reviewer’s comment. We have clarified and revised the sentence at Line 384-5 in the revised manuscript and as follows:
Discussion….The colorectal cancer-bearing mice model mimicked side effects of chemotherapy-associated gastrointestinal toxicity and diarrhea..….
Line 386 add the before intestinal
Response:We have add “the” before “intestinal” at Line 386 in the revised manuscript and as follows:
Discussion….Furthermore, the histological analysis of theintestinal mucosal injury indicated that the significant shortening of jejunal villi caused by FOLFOX in the mouse model was prevented by FMT..….
Line 422 reported inconsistently, correct
Response: Thank you for the reviewer’s comment. We have correct “reportedly inconsistent” to “reported inconsistently” at the Line 422 in the revised manuscript and as follows:
Discussion…FMT treatment upregulated FOLFOX-induced ZO-1 mRNA suppression in jejunum tissues. However, the effects of chemotherapeutic drugs such as 5-FU on the intestinal epithelial barrier via the expression of tight junction proteins are reported inconsistently..….
Add more references to line 430. There are more studies that should be acknowledged
Response: Thank you for the reviewer’s comment. We have added more reference (Wardill, H.R., et al., EBioMedicine 2019, 44, 730-740.) at Line 430 in the revised manuscript and as follows:
Discussion……Current studies on the safety of FMT in patients or animal models with tumors and in the context of treatment with anti-neoplastic agents are limited [13-16]. Cancer patients not only have a high mortality, because of individual episodes of sepsis, but suffer disproportionately from sepsis, as compared to the general population.…….
Line 434-5 in addition to chemotherapy, also cancer itself can cause bacterial translocation. Please add
Response: Thank you for the reviewer’s comment. In addition to chemotherapy, also cancer itself can cause bacterial translocation. We have add “cancer” at Line 434-5 in the revised manuscript and as follows:
Discussion….Gut pathogens are an important cause of sepsis in cancer patients. Damage to the gut epithelium after chemotherapy, cancer itselfand bacterial overgrowth contribute to bacterial translocation, making those receiving chemotherapy particularly vulnerable to bloodstream infections caused by enteric bacteria..….
Line 436-7, replace as by because and remove extracts
Response:We have replaced “as” by “because” and remove “extracts” at Line 436-7 in the revised manuscript and as follows:
Discussion….AsBecause feces extracts act as mediators between the donor and recipient, FMT has the potential to transmit occult infections, even when donor screening is performed stringently [13,14]...….
Line 422 (442?) add reference after pneumonia
Response: Thank you for the reviewer’s comment. We have added reference (Cammarota, G. et al., Gut 2017, 66, 569-580.) after pneumonia at Line 442 in the revised manuscript and as follows:
Discussion……mortality rate for Clostridium difficile infections, including septic shock, with decompensated toxic megacolon and fatal aspiration pneumonia [13]. Therefore, the safety of the FMT process continues to limit it.…….
Line 444 remove the microbiome-based therapeutic interventions
Response:We have removed “the microbiome-based therapeutic interventions” at Line 444 in the revised manuscript and as follows:
Discussion….Therefore, the safety of the FMT process continues to limit its use in immunocompromised and cancer patients treated with anti-neoplastic agents, and its long-term side effects, if any, remain unknown [13,14,16]. However, the microbiome-based therapeutic interventions of FMT might reportedly be able to correct dysbiosis and prevent bacterial translocation and sepsis, and improve survival....….
Line 446 add for instance before Li et al, correct FMT abbreviated
Response:We have added “for instance” before Li et al, and correct FMT abbreviated at Line 446 in the revised manuscript and as follows:
Discussion….For instance, Li et at showed that in a mouse model, fecal microbiota transplantationFMTcould not only reverse bacterial translocation, but also improve the survival period associated with experimental necrotizing enterocolitis [26].....….
Line 451 correct sentence related to bloodstream according to my comment above
Response:Thank you for the reviewer’s comment. We cannot state that no bacteria were translocated to the bloodstream in our manuscript at Line 451. We have clarified and revised the sentence at Line 451 as your suggestion in the revised manuscript and as follows:
Discussion….Our studies showed that in a colorectal cancer-bearing mouse model, at necropsy, none of the bacteria were detected to the bloodstream in all experimental groups;however, bacterial translocation to the mesenteric lymph nodes, liver, and spleen was more frequent in the FOLFOX-treated group than in the saline group.…….
Line 464 TLRs do not only recognize bacterial ligands but also viral and fungi. Please correct
Response: Thank you for reviewer’s comment. We have corrected it in the revised manuscript and as follows:
Discussion….TLRs recognized ligand from fungi, viruses and bacteria. Bacterial ligands that are not only unique to pathogens, but are found in all bacteria, and produced by symbiotic microorganisms [42].….
Line 475 it is not necessary to repeat the different cytokines in parentheses.
Response: Thank you for reviewer’s comment. We have corrected it in the revised manuscript and as follows:
Discussion….MyD88 is utilized by all TLRs and activates NF-κB and MAPKs, for the subsequent induction of inflammatory cytokine genes(, such as granulocyte colony-stimulating factor, interleukin-1β, IL-6, and TNF-α)..….
Add reference Line 483 references missing for the TLR ligands.
Response: Thank you for the reviewer’s comment. We have added references for the TLR ligands (Pekkala, S. et al., Cancers (Basel) 2019, 11.) at Line 483 in the revised manuscript and as follows:
Discussion……TLR4 recognizes bacterial lipopolysaccharide (LPS). TLR5 recognizes bacterial flagellin. The heterodimers formed by combinations of TLR1, TLR2, and 6 recognizes a wide variety of pathogen-associated molecular patterns, including lipoproteins, peptidoglycans, lipoteichoic acids, zymosan, mannan, and tGPI-mucin [22,23,45,48].In our study, FOLFOX treatment.…….
Correct that it is the heterodimers formed by combinations of TLR1, 2, & 6 that can achieve the recognition of so many ligands. For instance, TLR6 does not alone recognize peptidoglycans The paragraph dealing with cytokines should be combined or at least follow the paragraph of toll-like receptors. Both are linked
Response: Thank you for the reviewer’s comment. (1) It is the heterodimers formed by combinations of TLR1, 2, & 6 that can achieve the recognition of so many ligands. We have corrected it at Lind 483 in the revised manuscript and as follows. (2) The paragraph dealing with cytokines should be combined or at least follow the paragraph of toll-like receptors. Both are linked. We have corrected it at Lind 483 in the revised manuscript and as follows.
(1)Discussion……TLR4 recognizes bacterial lipopolysaccharide (LPS). TLR5 recognizes bacterial flagellin. The heterodimers formed by combinations of TLR1, TLR2, and 6 recognizes a wide variety of pathogen-associated molecular patterns, including lipoproteins, peptidoglycans, lipoteichoic acids, zymosan, mannan, and tGPI-mucin [22,23,44,47,48].. In our study, FOLFOX treatment…….
(2)Discussion……
TLRs act as a sensor for microbial infection, and are critical for the initiation of inflammation and immune defense responses [41]. ………Thus, our data suggest that FMT can suppress the effects of FOLFOX and induce the MyD88-dependent TLR signaling pathway; this indicates that intestinal microbiota of recipient mice in the FOLFOX group and homeostasis were reestablished.
The mechanisms by which the clinical benefits of FMT are achieved, including in inflammatory disease patients, are not completely understood. Several potential mechanisms …………Accordingly, our data suggest that by modulating the composition of the gut flora, FMT altered FOLFOX-induced changes in the gut microbiota and influenced the pathogenesis of mucositis via the gut microbiota-TLR-NF-kB signaling pathway in colorectal carcinoma-implanted mice..
Line 510 remove the sentence starting with Oral FMT. It is useless. Replace again as by because
Response: We have removed the sentence starting with Oral FMT. We have also replaced “as” by “because” at Line 510 in the revised manuscript and as follows:
Discussion….Oral FMT specifically caused a FOLFOX-induced change in gut microbiota, in colorectal carcinoma-implanted mice.Because a Jaccard analysis accounts merely for the presence/absence of OTUs and a Bray-Curtis analysis is more sensitive to changes in OTU abundance [55],..….
METHODS
Describe the animal procedures better. group size. Were they single-housed? If not, then the limitation is coprophagy > transferring the gut microbiota in the cage > the sample size has no power
Response:Thank you for the reviewers comment. The mice model for FMT study was based on our previous study (Chang, C.W et al., Front Microbiol 2018, 9, 983.),we reported on a colorectal cancer murine model with FOLFOX-induced intestinal mucositis that may enable us to effectively investigate the mechanism underlying intestinal injury, and its possible interaction with potential therapeutics.
In our study, mice (n=10 in each group) were co-housed in our rodent facility. Mice in the same group were co-housed.During cohousing, animals may feed on feces (also known as coprophagy) or ingest feces by self-grooming (Laukens, D. et al. FEMS Microbiol. Rev. 2016, 40, 117-132.). To avoid this factor possible, we keep the cage clean of feces during the study. Mice were housed in cages with wire mesh floors in order to let the feces fall through.
In this chemotherapy-induced intestinal toxicity animal model, studies with small size revealed FMT could safely ameliorate inflammation, protect the epithelium by maintaining intestinal epithelial integrity, and reduce the severity of mucositis following FOLFOX treatment. Base on the 3R principle (Replace, Reduce, Refine) for animal experimentation,we kept the number of animal experiments as low as possible and only use the necessary number of animals for study. Lastly, the distress inflicted upon the animals is as low as possible. We have added this limitation in the revised manuscript (Section: Materials and Methods, Discussion) and as follows:
Materials and Methods.
Animal experiments
…….Six to eight week old male BALB/c mice with a weight of 22–24 g were purchased from the National Laboratory Animal Center (Taipei, Taiwan) and co-housed in a rodent facilityat a temperature of 22 ± 1 °C, humidity of 55% ± 10%, and in a 12-h light-dark cycle ……..
Discussion…. Our study also has some limitations…..The sample size of mice was small and mice were co-housed, not single-housed for this study.We did not conduct an analysis of the time-course and dose-dependent effects of FMT;………..
The sections should be reorganized and redivided. Startying from the animals and cell cultures, followed by tumor implantation, FMT and chemotherapy. Body weight and other physiological measures should be an own chapter
Response:We have reorganized and re-divided the section of Material and Method (Section: 4.1-4.4). Starting from the animals and cell cultures, followed by tumor implantation, FMT and chemotherapy. Body weight and other physiological measures wer an own chapter in the revised manuscript (Section: Materials and Methods) as follow:
Materials and Methods…4.1 Animals.. 4.2 Cell culture…4.3 Tumor implantation 4.4 Transplantation of fecal microbiota…. 4.5 Chemotherapy regimen……4.6 Disease severity evaluation..….
chapter 4.5 the title should be Translocation of gut microbiota. The authors did not study infections
Response:We have changed the title of chapter 4.5 to “Translocation of gut microbiota” in the revised manuscript and as follows:
Method…4.5 Safety of FMT: Translocation of gut microbiota.….
Line 617 replace evaluation by analyses
Response:We have replaced “evaluation” by “analyses” at Line 617 in the revised manuscript and as follows:
Method…4.6 Histological evaluationanalyses.….
Line 621 remove wax
Response:We have removed “wax” at Lie 621 in the revised manuscript and as follows:
Method…and embedded in paraffin wax. Sections with a thickness of 4-μm were cut, mounted on glass slides, and stained with hematoxylin and eosin (H&E)[5].….
Line 629 replace we used somewhere by was used
Response:We have replaced we used somewhere by was used at Lie 629 in the revised manuscript and as follows:
Method…We used 10 mM sodium citrate (pH 6) or EDTA (pH 8) buffer was usedat 98 °C for 15 min to conduct heat-induced antigen retrieval and quenched the endogenous peroxidase activity using hydrogen peroxide for 10 min.….
chapter 4.9 start by RNA extraction, then cDNA synthesis then qPCR. Now it is mixed. In the two last sentences the same thing is said in two different ways.
Response:Thank you for the reviewer’s comment. We have clarified and revised the two last sentences in the Chapter 4.9 in the revised manuscript and as follows:
Method…
Relative expression of mRNA using real-time quantitative (q)PCR
RNA extractionfrom the jejunal specimens was ……….. Pairs of oligonucleotide primers specific to ZO-1, occludin, claudin-2, JAM-A, TLR1, TLR2, TLR4, TLR5, TLR6 and MyD88 are listed in Table 4. Gene expression was normalized to the GAPDH expression levels by using the following formula: ΔCt = (Ct of GAPDH−Ct of the gene).After setting the expression value of GAPDH to 1.0, the relative expression values were calculated by determining the 2 ΔCt value. The expression levels of ZO-1, occludin, claudin-2, JAM-A, TLR1, TLR2, TLR4, TLR5, TLR6, and MYD88 were analyzed using the comparative threshold cycle method (2−ΔΔCT), using GAPDH as an internal reference.….
Correct Line 671 do not use we collected
Response:We have removed “we collected somewhere” by “were collected “at Lie 671 in the revised manuscript and as follows:
Method…We collectedFreshfecal samples were collectedfrom mice and immediately stored …... We used the QIAmp® DNA stool mini kit (Qiagen, Germany) for the extraction of fecal DNA, according to the manufacturer’s instructions, and then stored isolated DNA.….
The authors should rationalize why the collected fecal sample and not colon content at necropsy. This is also a limitation. Colon content would be more represantitive of what is really happening inside the gut
Response:Thank you for the reviewer’s comment. The reasons why we collected fecal sample and not colon content at necropsy in this study;this FMT study was based on our previous study (Chang, C.W et al., Front Microbiol 2018, 9, 983.), using a colorectal cancer murine model with FOLFOX-induced intestinal mucositisfor further translational research in the clinical practice.Collected fecal sample would be more feasible. As reviewer’s comment, collect intestine or colon content would be more representative of what is really happening inside the gut. We have revised this in the revised manuscript (Section: Discussion) and as follows:
Discussion…. Our study also has some limitations.…….. Our microbiota studies were small size; only fecal sample, but not colon content, were collected and …....….
How long time DNA was stored at -30? It is not an optimal temperature
Response:Thank you for reviewer’s comment. Fresh fecal samples from mice and immediately stored at -80 °C. We corrected -30°C to -20°C and no freeze thaws for analysis in a short time. The reference (https://www.colorado.edu/ecenter/sites/default/files/attached-files/seracare_stability_of_genomic_dna_at_various_storage_conditions_isber2009.pdf) showed available temperature in various storage time. We clarified these in the revised manuscript and as follows:
Method…We used the QIAmp® DNA stool mini kit (Qiagen, Germany) for the extraction of fecal DNA, according to the manufacturer’s instructions, and then immediately isolated DNA at -20°C for analysis..….
Line 674 16S rRNA! There is no ribosomal DNA
Response:Thank you for reviewer’s comment. We have replaced “rDNA” by “rRNA “at Lie 674 in the revised manuscript and as follows:
Method…4.12.2 Sequencing and analysis of 16S rDNArRNA….
Remove thesentence strating at line 692 What was the rarefaction levels of the samples? How mny sequences? More information is needed Did the authors make FDR corrcetions. Please specify.
Response:Thank you for reviewer’s comment.(1)We have removed the sentence starting at Lind 692. in the revised manuscript and as follows. (2)The rarefaction levels of the samples and number of sequences analysis was based on the functions performed by a CLC bio "OTU clustering" (User manual for CLC Microbial Genomics Module, 2019,http://resources.qiagenbioinformatics.com/manuals/clcmgm/current/User_Manual.pdf). The algorithm filters out all OTUs whose combined abundance across all samples is less than the minimum combined abundance or whose combined abundance is less than the minimum combined abundance (% of all the reads) across all samples. The default value for the Minimum combined abundance is set at 10. In our analysis, rarefaction levels of the sample was 0, sequences average was 439,293 and sequence range was 2159,986~642,604. (3)Thanks for reviewer’s good suggestion. A PERMANOVA analysis for each pair of groups and the results of the test (pseudo-f statistic and p-value). Fisher’s Least Significant Difference (LSD) post hoc tests which correct for multiple testing were also shown. The false discovery rate (FDR) was a method of conceptualizing the rate of type I error in null hypothesis testing when conducting multiple comparisons. We wouldn’t use Bonferroni correction because we just used 5 mice. Bonferroni correction was very strict for our study, so we used LSD to control type I error. We have clarified and revised these in in the revised manuscript and as follows:
Method_4.12.2 Sequencing and analysis of 16S rRNA…. The pair-reads of sequences were merged into amplicon sequences using PEAR[75], and these amplicon sequences were processed to generate effective reads…CLC Genomics Workbench 12 software (CLC Bio, USA) was used along with the Greengenes 16S rRNA Taxonomy Database (gg_13_8) to conduct 16S rDNA analysis. “OTU clustering” function was used with de novo OTU clustering default parameter. “OTU clustering” function will cluster similar reads base on 97% similarity. The minimal criteria of OTU is 10 as default. The OTU clusters will be identified and used to following analysis and data visualization.…..
4.13 Statistical analysis: ……Taxonomy (i.e., phyla and OTUs) and α-diversity were analyzed by one-way ANOVA. The community structure (β-diversity) was analyzed by performing permutational multivariate analysis of variance (PERMANOVA) of ranked Bray-Curtis and Jaccard distances, using the CLC Genomics Workbench 12 software (CLC Bio, USA). A PERMANOVA analysis for each pair of groups and the results of the test (pseudo-f statistic and p-value). Fisher’s Least Significant Difference (LSD) post hoc tests which correct for multiple testing were also shown……
Chapter 4.11 should start with the program with which the analyses were made
Response:Thank you for the reviewer’s comment. We should start with the program with which the analyses were made in Chapter 4.11. We have clarified and revised these in in the revised manuscript and as follows:
4.13 Statistical analysis:
Results are presented as the mean ± standard error of the mean (SEM). ………. The community structure (β-diversity) was analyzed by performing permutational multivariate analysis of variance (PERMANOVA) of ranked Bray-Curtis and Jaccard distances, using the CLC Genomics Workbench 12 software (CLC Bio, USA). A PERMANOVA analysis for each pair of groups and the results of the test (pseudo-f statistic and p-value). Fisher’s Least Significant Difference (LSD) post hoc tests which correct for multiple testing were also shown. The results were considered to be statistically significant if P < 0.05.
CONSCLUSIONS
the authors draw too bold conclusions. Please modify the sentences. In addition, use past time in the verbs. Remove NFKB activation as commented above.
Response:Thank you for reviewer’s comment. We have modified the sentences in Conclusions in the revised manuscript and as follows:
Conclusions…Our murine model of colorectal cancer with severe FOLFOX-induced intestinal mucositis exhibited changes in gut microbiota; the development of mucositis might be encouraged by the activation of the gut microbiota present downstream of the TLR signaling pathway, following MyD88 and NF-κB expression. NF-kB expression results in the generation of apoptotic signals and pro-inflammatory cytokines, which sequentially contribute to intestinal mucosal integrity. Through the modulation of the gut microbiota present downstream of the TLR signaling pathway and by generating pro-inflammatory responses, FMT mitigated FOLFOX-induced intestinal toxicity in a safe manner. FMT could safely ameliorate inflammation, protect the epithelium by maintaining intestinal epithelial integrity, and reduce the severity of mucositis following FOLFOX treatment. The possible mechanisms may involve the gut microbiota-TLR-MyD88-NF-kB signaling pathway in mice with implanted colorectal carcinoma cells. Specifically, FMT is able to heighten the survival rate of chemotherapy-treated mice and could be used in a novel therapeutic strategy for managing chemotherapy-induced mucositis, to improve the prognosis of patients receiving chemotherapy in the future..
Round 2
Reviewer 2 Report
The authors have managed to review and correct most of my suggestions. However, there are still errors that need to be corrected.
The introduction still does not describe what are gut microbes and what they do. This is a nutritional journal, not microbial, and for all readers the gut microbes may not be familiar. Title 2.6.: The effects of FMT on the regulation of expression of serum inflammatory cytokines in colorectal cancer-401 bearing mice challenged with FOLFOX. REPLACE regulation of expression BY levels The authors are still unable to interpret their gut microbiota heat map results, which may be due to that they do not know what the heat map shows. The heat map shows that at phylum levels the microbiota of saline and FOLFOX + FMT cluster together resembling more distantly FMT. According to heat map FOLFOX group is the most different group The authors need to add the results at least at genus level somewhere. Genus is much more important than the broad category of Bacteroidetes and Firmicutes. Were there significant differences at genus level and which? Genus level abundances can be easily drawn with CLC package, and there is space in the figure to include those. It is a great limitation that each group of mice were co-housed. Reagarding gut microbiota studies it really makes the n=1 in each group. Coprophagy and transmission of gut microbiota between littermates is a well known fact. This is a limitation even if the authors state: “keep the cage clean of feces during the study. Mice were housed in cages with wire mesh floors in order to let the feces fall through.” This limitation should be more clearly written, and the authors should state that the study needs to be repeated in different groups of mice before FMT can be recommended for clinical trials in chemotherapy patients. Therefore the last sentence in the conclusion is very bold and should be modified -20°C is not an optimal storage temperature for DNA, and it should not go through repetitive freeze/thaw cycles but should be divided in several tubes upon isolation and before freezing I still not see in the methods for which analyses the authors did this: “The false discovery rate (FDR) was a method of conceptualizing the rate of type I error in null hypothesis testing when conducting multiple comparisons”. In addition, Was the false discovery rate (FDR) Benjamini-Hochberg? Did the authors subsample their data? The authors state that “In our analysis, rarefaction levels of the sample was 0”. When comparing the diversity values of different samples, they should look the values based on equal read numbers. Or, they can also subsample even number of reads for each sample, one by one (or in batch) after the first trimming step in CLC package: Utility tools – Sample Reads. Sample an absolute number (select the number of your smallest read numbers), sample type (random). And then you should make the OTU clustering etc. for resampled samples, and do the diversity analyses Did the authors trim their sequences and did they filter out low count OTUs? Which was set as low count? In the text I may understand that it was 10, but 10 still quite low Which statistical test did the authors use for differential abundance analysis? Microbiome data is hardly ever normally distributed, so in the analysis the authors should use Kruskall-Wallis or Mann Whitney. Both options are available in CLC package. Which distance was used to produce the heat map, Euclidean or which? There are few options in CLC
Author Response
December 18, 2019
Editor-in-Chief
International Journal of Molecular Sciences
Dear Editor and Reviewers:
Thank you for your letter and comments on our manuscript titled “Fecal microbiota transplantation prevents intestinal injury and Toll-like receptors upregulation from 5-fluorouracil/oxaliplatin toxicity in colorectal cancer” (ID: 651973). The paper was co-authored by Ching-Wei Chang, Hung-Chang Lee, Li-Hui Lee, Jen-Shiu Chiang Chiau, Tsang-En Wang, Wei-Hung Chuang, Ming-Jen Chen, Horng-Yuan Wang, Shou-Chuan Shih, Chia-Yuan Liu, and Tung-Hu Tsai. All comments were insightful and helped to considerably improve our manuscript. We have carefully addressed each issue raised by the reviewers, and the point-by-point responses to their comments are attached.
Intestinal mucositis is a common adverse effect associated with the use of FOLFOX for treating patients with colorectal cancer. Its occurrence results in increased hospitalization duration and infection risk and reduced levels of anti-neoplastic agents used for treatment. This contributes to a reduced survival rate and a substantial burden on Medicare. Thus, we developed a convenient and novel method for alleviating mucositis by investigating the effects of a fecal microbiota transplant (FMT) on FOLFOX-induced mucosal injury in BALB/c mice implanted with syngeneic CT26 colorectal adenocarcinoma cells.We believe that our study makes a significant contribution to the literature because our results show that FMT safely reduces the severity of intestinal mucositis and diarrhea, without affecting the anti-tumor effects of FOLFOX or causing bacteremia in colorectal cancer-bearing mice.
Further, we believe that this paper will be of interest to the readership of your journal because our results demonstrate that FMT can mitigate FOLFOX-induced intestinal toxicity in a safe manner. Specifically, FMT can increase the survival rate of chemotherapy-treated mice and can be used as a novel therapeutic strategy for managing chemotherapy-induced mucositis to improve the prognosis of patients with colorectal cancer patients undergoing chemotherapy.
This manuscript has not been published or presented elsewhere in part or in entirety and is not under consideration by another journal. The study design was approved by the appropriate ethics review board. We have read and understood your journal’s policies, and we believe that neither the manuscript nor the study violates any of these.There are no conflicts of interest to declare.
Thank you for your consideration. I look forward to hearing from you.
Sincerely,
Yu-Jen Chen M.D., Ph.D.
Department of Radiation Oncology, Mackay Memorial Hospital,
No. 92, Section 2, Chung San North Road, Taipei 104, Taiwan
Fax: (886) 2 2809 6180
Phone: (886) 2 2809 4661 ext. 2301
E-mail: chenmdphd@gmail.com
Point-by-point responses to the reviewers’ comments (Chen et al.)
Title: Fecal microbiota transplantation prevents intestinal injury and Toll-like receptors upregulation from 5-fluorouracil/oxaliplatin toxicity in colorectal cancer
We sincerely thank the reviewers for the constructive criticisms and valuable comments, which helped to considerably improve the manuscript. Our responses to the reviewers’ comments are provided below.
Reviewers' comments:
Reviewer #2:
Q1. The introduction still does not describe what are gut microbes and what they do.This is a nutritional journal, not microbial, and for all readers the gut microbes may not be familiar.
Response:
Thank you for the reviewer’s comments. We have described gut microbes in more detail in the Introduction section of the revised manuscript as follows:
Introduction
The microbiota, formed by microorganisms, residing in the gastrointestinal tract, is referred to as the ‘intestinal microbiota’ or ‘gut microbiota’ (Aarnoutse, R et al., 2019, Alexander, J.L. et al., 2017). The microbiota affects various aspects of human health, including providing nutrients and vitamins, protecting against pathogens, epithelial mucosa homeostasis, and immune system development (Villeger, R et al., 2019). Microbial dysbiosis has been linked to various metabolic and inflammatory diseases, such as diabetes mellitus, hypertension, inflammatory bowel disease, and obesity (Aarnoutse, R et al., 2019, Alexander, J.L. et al., 2017). Growing evidence not only implies that chemotherapeutics affect the intestinal microbial composition, but also that multidirectional interactions between the gut microbiota and host immune system may influence the development and progression of chemotherapy-induced intestinal inflammation [4-6].
Reference:
Aarnoutse, R.; Ziemons, J.; Penders, J.; Rensen, S.S.; de Vos-Geelen, J.; Smidt, M.L. The Clinical Link between Human Intestinal Microbiota and Systemic Cancer Therapy. Int J Mol Sci 2019, 20.
Villeger, R.; Lopes, A.; Carrier, G.; Veziant, J.; Billard, E.; Barnich, N.; Gagniere, J.; Vazeille, E.; Bonnet, M. Intestinal Microbiota: A Novel Target to Improve Anti-Tumor Treatment? Int J Mol Sci 2019, 20
Alexander, J.L.; Wilson, I.D.; Teare, J.; Marchesi, J.R.; Nicholson, J.K.; Kinross, J.M. Gut microbiota modulation of chemotherapy efficacy and toxicity. Nat Rev Gastroenterol Hepatol 2017, 14, 356-365.
Cani, P.D. Human gut microbiome: hopes, threats and promises. Gut 2018, 67, 1716-1725
Q2. Title 2.6.: The effects of FMT on the regulation of expression of serum inflammatory cytokines in colorectal cancer-401 bearing mice challenged with FOLFOX. REPLACE regulation of expression BY levels
Response:
Thank you for your comments. We have replaced “regulation of expression” with “levels” in the revised manuscript as follows:
Results….2.6 The effects of FMT on the regulation of expression levelsof serum inflammatory cytokines in colorectal cancer-bearing mice challenged with FOLFOX
Q3. The authors are still unable tointerpret their gut microbiota heat map results, which may be due to that they do not know what the heat map shows. The heat map shows that at phylum levels the microbiota of saline and FOLFOX + FMT cluster together resembling more distantly FMT. According to heat map FOLFOX group is the most different group The authors need to add the results at least at genus level somewhere. Genus is much more important than the broad category of Bacteroidetes and Firmicutes. Were there significant differences at genus level and which? Genus level abundances can be easily drawn with CLC package, and there is space in the figure to include those.
Response:
Thank you for these comments. We will respond to each comment individually:
(1) We apologize that our response to the comment regarding the heatmap results was not well-explained.In our study, the heatmap at the phylum levels revealed that the saline and FMT + FOLFOX groups clustered together, resembling a more distantly FMT group. The FOLFOX group showed the largest differences. Accordingly, FOLFOX altered the composition of the gut microbiota. FMT altered this composition in the FOLFOX-challenged group, as compared to that in the saline group. We have revised this in our revised manuscript as follows.
(2) We have analyzed the genus level abundances drawn with the CLC package. The results are shown in bar charts and a heatmap for the relative abundance of the gut microbiota at the genus level. The taxonomic analysis results at the genus level revealed that FOLFOX changed the composition of the gut microbiota. FMT altered this composition in the FOLFOX-challenged group compared to in the saline group(figure of bar charts and heatmap as follows). The heatmap at the genus level revealed that saline and FMT groups clustered together, resembling a more distantly FOLFOX and FMT + FOLFOX groups (figure of heatmap as follows).In differential abundance analysis at the genus level, we list the significant differences of gut microbiota in the Table as follows. We have revised these points in our revised manuscript (Results section and figure) and as follows.
Figure The composition of gut microbiota was determined by the following: Bar charts, Heatmap of the relative abundance of gut microbiota at the genus level
Table. Significant differences of fecal gut microbiota at genus level in differential abundance analysis
|
Name |
Max group mean (OTU) |
P-value |
FDR p-value |
|
{Unknown Genus} Coriobacteriaceae |
103.40 |
0.000 |
0.000 |
|
{Unknown Family} Bacteroidales |
8305.60 |
0.000 |
0.000 |
|
Odoribacter |
335.60 |
0.000 |
0.001 |
|
Parabacteroides |
1459.60 |
0.000 |
0.000 |
|
Prevotella |
1406.40 |
0.000 |
0.001 |
|
{Unknown Genus} Rikenellaceae |
5265.60 |
0.000 |
0.002 |
|
{Unknown Genus} S24-7 |
40784.80 |
0.000 |
0.000 |
|
{Unknown Family} YS2 |
133.20 |
0.000 |
0.000 |
|
{Unknown Family} Clostridiales-2 |
16182.80 |
0.000 |
0.000 |
|
Clostridium-2 |
206.60 |
0.000 |
0.000 |
|
Roseburia |
48.60 |
0.000 |
0.000 |
|
{Unknown Genus} Peptostreptococcaceae |
138.20 |
0.000 |
0.000 |
|
{Unknown Genus} Ruminococcaceae-1 |
26.00 |
0.000 |
0.000 |
|
Anaerotruncus |
30.40 |
0.000 |
0.001 |
|
{Unknown Genus} Erysipelotrichaceae-1 |
124.00 |
0.000 |
0.001 |
|
Allobaculum |
215.40 |
0.000 |
0.000 |
|
Sutterella |
1102.40 |
0.000 |
0.001 |
|
{Unknown Genus} Desulfovibrionaceae |
1928.80 |
0.000 |
0.001 |
|
{Unknown Genus} Helicobacteraceae-2 |
2166.60 |
0.000 |
0.000 |
|
Helicobacter |
1495.00 |
0.000 |
0.000 |
|
Shigella |
1400.00 |
0.000 |
0.000 |
|
Haemophilus |
88.20 |
0.000 |
0.000 |
|
Anaeroplasma |
15.00 |
0.000 |
0.000 |
|
Akkermansia |
1634.60 |
0.000 |
0.000 |
|
N/A |
22.80 |
0.000 |
0.000 |
FDR: false discovery rate. If P < 0.05, differences were considered significant.
Result
2.7 The effects of FMT ……colorectal cancer-bearing mice
The composition of gut microbiota……. Taxonomic analysis at the phylum level revealed that FOLFOX changed the composition of the gut microbiota. FMT altered this composition in the FOLFOX-challenged group, as compared to that in the saline group (Figure 7A and 7B).The heatmap at the phylum levels revealed that the saline and FMT + FOLFOX groups clustered together, resembling a more distantly FMT group. The FOLFOX group is the most different group (Figure 7B). Taxonomic analysis at the family and genus levels also showed the same trends of the FMT effects (data not shown)…….Taxonomic analysis at the genus level also revealed that FOLFOX altered the composition of the gut microbiota. FMT altered this composition in the FOLFOX-challenged group, as compared to that in the saline group (Figure 7C and 7D). The heatmap at the genus level revealed that the saline and FMT groups clustered together, resembling a more distantly FOLFOX and FMT + FOLFOX groups (Figure 7D). Bacteroidetes (B) and Firmicutes (F) were identified as the major phyla present in all groups…………………………….
Figure.Changes in fecal gut microbiota of subcutaneously injected colorectal cancer mice presenting FOLFOX-induced intestinal damage with/without a FMT (n = 5 for each group). The composition of gut microbiota was determined by the following: (A) Bar charts, (B) Heatmap of the relative abundance of gut microbiota at the phylum level and (C) Bar charts, (D) Heatmap at the genus level, (E)Abundance of Firmicutes-to-Bacteroidetes (F/B) ratio and (F)Tukey box plots of α-diversity. FMT50: 50 mg/ml. Values are presented as the mean ± SEM. *: p< 0.05, **: p< 0.01, ***: p< 0.001
Q4. It is a great limitation that each group of mice were co-housed. Reagarding gut microbiota studies it really makes the n=1 in each group. Coprophagy and transmission of gut microbiota between littermates is a well known fact. This is a limitation even if the authors state: “keep the cage clean of feces during the study. Mice were housed in cages with wire mesh floors in order to let the feces fall through.” This limitation should be more clearly written,andthe authors should state that the study needs to be repeated in different groups of mice before FMT can be recommended for clinical trials in chemotherapy patients.Therefore the last sentence in the conclusion is very bold and should be modified
Response:
Thank you for the your comment and for pointing the limitations of our study. Coprophagy and transmission of gut microbiota between littermates are well-known phenomena(Laukens, D. et al. FEMS Microbiol. Rev. 2016, 40, 117-132.). Accordingly, we have clarified this information in the revised manuscript in the Discussion section as follows. We have also stated that“the study needs to be repeated and validated in different groups of mice single-housed in the future before clinical trials of FMT.” in the revised discussion section. In addition, we have modified the last sentence in the conclusion of the revised manuscript. The following changes were made in the revised manuscript:
Discussion
Our study also has some limitations. We did not work……………mice were co-housed, not single-housed for this study. During cohousing, animals may feed on feces (also known as coprophagy) or ingest feces by self- grooming (Laukens, D et al., 2016). It will be a risk of transmission of gut microbiota between littermates (Laukens, D et al., 2016). The study may need to be repeated and validated in different groups of mice single-housed in the future before clinical trials of FMT.……………………..
Conclusions
FMT could safely ameliorate inflammation, ……………. in mice with implanted colorectal carcinoma cells. FMT may improve the survival rate of chemotherapy-treated mice. Different groups of mice should be analyzed before FMT can be recommended for clinical trials in cancer patients undergoing chemotherapy.
References:
Laukens, D.; Brinkman, B.M.; Raes, J.; De Vos, M.; Vandenabeele, P. Heterogeneity of the gut microbiome in mice: guidelines for optimizing experimental design. FEMS Microbiol. Rev. 2016, 40, 117-132.
Q5. -20°Cis not an optimal storage temperature for DNA, and it should not go through repetitive freeze/thaw cycles but should be divided in several tubes upon isolation and before freezing
Response:
Thank you for the reviewer’s comment. We used the QIAmp® DNA stool mini kit (Qiagen, Germany) to extract of fecal DNA. We stored the extracted DNA at -20°C with dividing the samples into several tubes upon isolation and before freezing according to the manufacturer's instruction (QIAamp DNA & Stool Handbook 06/2012) and previously published papers (Deng, H. et al., Front Microbiol 2018, 9, 2976; Jiang, Z.D.., Anaerobe 2017, 48, 110-114.; Chang, C.W., Front Microbiol 2018, 9, 983.). We did not perform repetitive freeze/thaw cycles of the extracted DNA. Accordingly, we have clarified and revised this point in the revised manuscript and as follows:
Method…
4.12.1 Sample collection and DNA extraction
Fresh fecal samples were collected from mice and immediately stored at -80 °C. We used the DNA from fecal materials was extracted by using a QIAmp® DNA stool mini kit (Qiagen, Germany). We stored the extracted DNA at -20°C with dividing it into several tubes upon isolation and before freezing for the extraction of fecal DNA, following the manufacturer’s instructions without performing repetitively freeze/thaw cycles. ,and then immediately isolated DNA at -20°C for analysis..…
Reference:
QIAamp®DNA Stool Handbook (for DNA purification from stool samples) 2ndedition 06/2012
Chang, C.W.; Liu, C.Y.; Lee, H.C.; Huang, Y.H.; Li, L.H.; Chiau, J.C.; Wang, T.E.; Chu, C.H.; Shih, S.C.; Tsai, T.H., et al. Lactobacillus casei Variety rhamnosus Probiotic Preventively Attenuates 5-Fluorouracil/Oxaliplatin-Induced Intestinal Injury in a Syngeneic Colorectal Cancer Model. Front Microbiol 2018, 9, 983.
Deng, H.; Yang, S.; Zhang, Y.; Qian, K.; Zhang, Z.; Liu, Y.; Wang, Y.; Bai, Y.; Fan, H.; Zhao, X., et al. Bacteroides fragilis Prevents Clostridium difficile Infection in a Mouse Model by Restoring Gut Barrier and Microbiome Regulation. Front Microbiol 2018, 9, 2976.
Jiang, Z.D.; Alexander, A.; Ke, S.; Valilis, E.M.; Hu, S.; Li, B.; DuPont, H.L. Stability and efficacy of frozen and lyophilized fecal microbiota transplant (FMT) product in a mouse model of Clostridium difficile infection (CDI). Anaerobe 2017, 48, 110-114.
Q6. I still not see in the methods for which analyses the authors did this: “The false discovery rate (FDR) was a method of conceptualizing the rate of type I error in null hypothesis testing when conducting multiple comparisons”. In addition, Was the false discovery rate (FDR) Benjamini-Hochberg? Did the authors subsample their data?
Response:
Thank you for the reviewer’s comment.. The false discovery rate (FDR) Benjamini & Hochberg procedure (Benjamini & Hochberg, 1995)is a powerful analysis method that decreases the FDR. Adjusting this rate helps to control for the fact that sometimes small p-values (less than 0.05) happen by chance, which could lead us to incorrectly reject the true null hypotheses. Thus, the B-H procedure helps to avoid Type I errors (false-positives). The FDRs were calculated using R software (version 3.6.0 R package “stats”, function “p.adjust”, correction ("BH" or its alias "fdr")).
We have used the false discovery rate (FDR) (Benjamini & Hochberg, 1995) for subsampling and to conduct multiple comparisons of our data. We observed that significant differences in the beta diversity of the microbiota of the saline, FMT, FOLFOX, and FOLFOX+FMT groups (p < 0.05; FDR corrected p < 0.05); FMT and FOLFOX treatment alone affected the composition of the fecal gut microbiota. Furthermore, the oral administration of FMT specifically altered the FOLFOX-affected gut microbiota in colorectal carcinoma-implanted mice (p < 0.05; FDR corrected p < 0.05).We have revised this point in the revised manuscript (Table and Results ) and as follows:
Results:
Beta diversity was analyzed using the Jaccard and Bray-Curtis distance matrix analysis method, as illustrated in Table 3, and it showed that there were significant differences in the beta diversity of the microbiota of the saline, FMT, FOLFOX, and FOLFOX+FMT groups (p< 0.05; FDR correctedp< 0.05 ); FMT and FOLFOXtreatment alone affected the composition of fecal gut microbiota. Furthermore, the oral administration of FMT specifically altered FOLFOX-affected gut microbiota in colorectal carcinoma-implanted mice(p < 0.01; FDR corrected p< 0.05).
Method:
Comparisons between more than three groups were made using the One-way ANOVA. Data were analyzed using IBM SPSS software (version 21.0; SPSS Institute).Taxonomy (i.e., phyla and OTUs) and α-diversity were analyzed………….the CLC Genomics Workbench 12 software (CLC Bio, USA). A PERMANOVA analysis for each pair of groups and the results of the test (pseudo-f statistic and p-value). False discovery rate (FDR) (Benjamini & Hochberg, 1995) which correct for multiple testing were also shown. The FDRs were calculated using R software (version 3.6.0 R package “stats”, function “p.adjust”, correction ("BH" or its alias "fdr")).The results were considered to be statistically significant if P < 0.05.
Table P, F values and FDR correction from tests of Bray-Curtis and Jaccard similarity indices to evaluate the significance of FMT in changing the gut microbiota from the stool of subcutaneously injected colorectal cancer mice challenged with FOLFOX (n=5 for each group)
|
|
|
Bray-Curtis |
|
|
Jaccard |
|
||
|
|
F |
P |
FDR corrected p values |
|
F |
P |
FDR corrected p values |
|
|
Saline |
FMT50 |
2.15145 |
0.00794 |
0.009528 |
|
1.88344 |
0.00794 |
0.011910 |
|
Saline |
FOLFOX |
2.94918 |
0.03968 |
0.039680 |
|
2.37435 |
0.03968 |
0.039680 |
|
FMT50 |
FOLFOX |
4.16423 |
0.00794 |
0.009528 |
|
2.94634 |
0.00794 |
0.011910 |
|
Saline |
FOLFOX + FMT50 |
8.49609 |
0.00794 |
0.009528 |
|
5.64205 |
0.00794 |
0.011910 |
|
FMT50 |
FOLFOX + FMT50 |
10.77795 |
0.00794 |
0.009528 |
|
6.44849 |
0.00794 |
0.011910 |
|
FOLFOX |
FOLFOX + FMT50 |
2.73125 |
0.00794 |
0.009528 |
|
2.35479 |
0.01587 |
0.019044 |
FMT50: 50 mg/ml. FDR: false discovery rate.If P < 0.05, differences were considered significant.
Reference:
Benjamini Y, Hochberg Y. 1995. Controlling the false discovery rate: a practical and powerful approach to multiple testing. Journal of the Royal Statistical Society Serier B-Statistical Methodology 57 (1): 289-300.
R Core Team. R: a Language and Environment for Statistical Computing; R Foundation for Statistical Computing: Vienna, Austria, 2019. https://www.R-project.org/.
Q7. The authors state that “In our analysis, rarefaction levelsof the sample was 0”. When comparing the diversity values of different samples, they should look the values based on equal read numbers. Or,they can also subsample even number of reads for each sample, one by one (or in batch) after the first trimming step in CLC package: Utility tools – Sample Reads. Sample an absolute number (select the number of yoursmallest read numbers), sample type (random). And then you should make the OTU clustering etc. for resampled samples, and do the diversity analysesDid the authors trim their sequencesand did they filter out low count OTUs? Which was set as low count? In the text I may understand that it was 10, but 10 still quite low
Response:
Thank you for your comments. We would like to respond to point by point.
(1) We corrected that our rarefaction levels is not 0 but according to the CLC setting. Our rarefaction analysis in CLC is based on presets (http://resources.qiagenbioinformatics.com/manuals/clcmgm/current/User_Manual.pdf,Page 58, CLC tools: Metagenomics >> Abundance Analysis >> Alpha Diversity). The rarefaction analyses in CLC "Alpha Diversity" function are done differently depending on the type of abundance table used as input.
For OTU tables, where abundances are counts, rarefaction is calculated by sub-sampling the abundances in the different samples at different depths. Following are the default parameters of rarefaction analysis:
Minimum depth to sample = 1
Numbers of points = 20
Replicates at each depth = 100
(2)We do not used the “Trim Reads” function in the workflow of CLC Microbial Genomics Module.
(3)Generally, we did not trim any reads. However, we filtered the low count OTUs which are lower than 10 before doing our figures and statistics. The pre-filter OTU table is a sparse table, much of OTU is near 0 in each sample. the value is lower, we picked up the result were greater than this.
Q8.Which statistical testdid the authors use fordifferential abundance analysis? Microbiome data is hardly ever normally distributed, so in the analysis the authors should use Kruskall-Wallisor Mann Whitney. Both options are available in CLC package. Which distancewas used to produce the heat map, Euclidean or which? There are few options in CLC
Response:
Thank you for the reviewer’s comments. We would like respond to point by point.
(1) According to the manual in CLC Microbial Genomics Module (http://resources.qiagenbioinformatics.com/manuals/clcmgm/current/User_Manual.pdf, page 65), the differential abundance analysis is carried by “Generalized Linear Model”, GLM, where is assumed that abundances follow a Negative Binomial distribution.
(2)Microbiome data is hardly ever normally distributed, so we use one-way Kruskal-Wallis H test rather than one-way ANOVA test to evaluate alpha diversity. This test is a nonparametric alternative to ANOVA (i.e., it does not depend on the data following a given distribution - the normal distribution in case of ANOVA), and followed by Mann-Whitney U post hoc test. Mann-Whitney U test to specifically determine which pairs of groups follow different distributions. Our study showed that the alpha diversity indexes, including the Shannon entropy (community diversity) were determined, and no significant differences were observed in the microbiota of the saline, FMT, FOLFOX, and FOLFOX+FMT group (figure as follow).
(3)We used Manhattan distance to produce the heatmap in CLC.
We have revised these points in the revised manuscript (Section: Figures, Results, Statistical analysis) and as follows.
Figure Changes in fecal gut microbiota of subcutaneously injected colorectal cancer mice presenting FOLFOX-induced intestinal damage with/without a FMT (n = 5 for each group). The composition of gut microbiota was determined by Tukey box plots of α-diversity. FMT50: 50 mg/ml. Values are presented as the mean ± SEM. *: p< 0.05, **: p < 0.01, ***: p < 0.001.
Results:
………. The alpha diversity indexes, including the Shannon entropy (community diversity) were determined, and no significant differences were observed in the microbiota of the saline, FMT, FOLFOX, and FOLFOX+FMT groups,…….Beta diversity was analyzed using the Jaccard……..
Statistical analysis
Results are presented as the mean ± standard error of the mean (SEM). ………….Taxonomy (i.e., phyla and OTUs) and α-diversity were analyzed by one-way Kruskal-Wallis H test and followed by Mann-Whitney U post hoc test. The community structure (β-diversity) was analyzed …. ……………...

Round 3
Reviewer 2 Report
I congratulate the authors for assessing and correcting all my concerns. I have no further comments.